# Rethinking Diffusion Posterior Sampling: From Conditional Score Estimator to Maximizing a Posterior

**Tongda Xu**[1,2]**, Xiyan Cai**[1,3]**, Xinjie Zhang**[4]**, Xingtong Ge**[5]**, Dailan He**[6]**,**
[1]Institute for AI Industry Research, Tsinghua University
[2]Department of Computer Science and Technology, Tsinghua University

**Ming Sun**[7]**, Jingjing Liu**[1]**, Ya-Qin Zhang**[1]**, Jian Li**[8*]**, Yan Wang**[1*]
[3]New York University, [4]Hong Kong University of Science and Technology, [5]SenseTime Research
[6]The Chinese University of Hong Kong, [7]Kuaishou Technology
[8]Institute for Interdisciplinary Information Sciences, Tsinghua University
x.tongda@nyu.edu, lapordge@gmail.com, wangyan@air.tsinghua.edu.cn

## Abstract

Recent advancements in diffusion models have been leveraged to address inverse problems without additional training, and Diffusion Posterior Sampling (DPS) (Chung et al., 2022a) is among the most popular approaches. Previous analyses suggest that DPS accomplishes posterior sampling by approximating the conditional score. While in this paper, we demonstrate that the conditional score approximation employed by DPS is not as effective as previously assumed, but rather aligns more closely with the principle of maximizing a posterior (MAP). This assertion is substantiated through an examination of DPS on $512 \times 512$ ImageNet images, revealing that: 1) DPS's conditional score estimation significantly diverges from the score of a well-trained conditional diffusion model and is even inferior to the unconditional score; 2) The mean of DPS's conditional score estimation deviates significantly from zero, rendering it an invalid score estimation; 3) DPS generates high-quality samples with significantly lower diversity. In light of the above findings, we posit that DPS more closely resembles MAP than a conditional score estimator, and accordingly propose the following enhancements to DPS: 1) we explicitly maximize the posterior through multi-step gradient ascent and projection; 2) we utilize a light-weighted conditional score estimator trained with only 100 images and 8 GPU hours. Extensive experimental results indicate that these proposed improvements significantly enhance DPS's performance. The source code for these improvements is provided in this link.

## 1 Introduction

In recent years, diffusion models have emerged as a powerful tool for solving inverse problems without additional training (Song et al., 2020; Chung et al., 2022a; Rout et al., 2024). One prominent approach leveraging diffusion models to tackle inverse problems is Diffusion Posterior Sampling (DPS) (Chung et al., 2022a). DPS has gained significant attention as it effectively produce high quality samples for various image restoration problems, such as super resolution and deblurring.

The conventional understanding of DPS posits that it approximates the conditional score to achieve posterior sampling (Chung et al., 2022a; Song et al., 2023c). Various subsequent works follow this explanation (Yu et al., 2023; Boys et al., 2023; Rout et al., 2023; Chung et al., 2023; Yang et al., 2024). However, recent theoretical study has revealed that this approximation has a large error lower bound (Yang et al., 2024). In this paper, we challenge the prevailing view by presenting a numerical examination of DPS in practical scenarios, particularly for $512 \times 512$ ImageNet images. Our analysis

---

[*]To whom the correspondence should be addressed.

reveals that DPS aligns more closely with the principles of maximizing a posterior (MAP) rather than conditional score estimation.

From our empirical study, we make three major observations: 1). The conditional score estimation of DPS considerably diverges from the score of a properly trained conditional diffusion model and is even outperformed by an unconditional score; 2). The mean of DPS's conditional score estimate significantly deviates from zero, thus failing to qualify as a valid score; 3). The samples generated by DPS, although of high quality, exhibit markedly lower diversity. These findings collectively argue that DPS more closely aligns with the MAP framework than with conditional score estimation.

Given these insights, we propose enhancements to DPS to better align it with the concept of MAP. Our proposed modifications include: 1) Explicit maximization of the posterior through multi-step gradient ascent and projection; 2) The employment of a lightweight conditional score estimator trained with 100 images and 8 GPU hours. Extensive experimental evaluations demonstrate that these improvements notably boost the performance of DPS.

Our technical contributions can be summarized as follows:

- (Section 3) We demonstrate that DPS aligns more closely with the principles of maximizing a posterior (MAP) than the conditional score estimation, by showing it exhibits significant score estimation errors, a high score mean, and low sample diversity.
- (Section 4) We introduced a multi-step gradient ascent algorithm to explicitly maximize the posterior and a lightweight conditional score estimator trained with 100 images and 8 GPU hours, and both of them significantly boost DPS's performance.
- (Section 5) Our extensive experimental results substantiate the effectiveness of the proposed enhancements, significantly improving the performance metrics of DPS.

## 2 PRELIMINARIES

**Denoising Diffusion Probability Model** Diffusion models represent an important class of generative models, which utilizes a $T$-step Gaussian Markov chain (Sohl-Dickstein et al., 2015). One of the most widely adopted diffusion models is the Denoising Diffusion Probability Model (DDPM) (Ho et al., 2020). We denote the source image as $X_0$, and the forward Markov chain of DDPM is:

$$q(X_T, ..., X_1|X_0) = \prod_{t=1}^{T} q(X_t|X_{t-1}), \text{ where } q(X_t|X_{t-1}) = \mathcal{N}(\sqrt{1-\beta_t}X_{t-1}, \beta_t I), \quad (1)$$

where $\beta_t$ are hyperparameters. The reverse process of DDPM is a Markov chain with the transition $p_\theta(X_{t-1}|X_t)$, with the score function $\nabla_{X_t} \log p(X_t)$ approximated by a neural network $s_\theta(X_t, t)$:

$$p_\theta(X_0, ..., X_T) = p(X_T) \prod_{t=1}^{T} p_\theta(X_{t-1}|X_t),$$

$$\text{where } p_\theta(X_{t-1}|X_t) = \mathcal{N}(\frac{1}{\sqrt{\alpha_t}}(X_t + \beta_t s_\theta(X_t, t)), \sigma_t^2 I), \quad (2)$$

and $\alpha_t, \sigma_t^2$ are parameters determined by $\beta_t$ (Ho et al., 2020).

**Diffusion Posterior Sampling** On the other hand, Diffusion Posterior Sampling (DPS) (Chung et al., 2022a) extends DDPM by enabling conditional sampling. Given an operator $f(.)$ and an observation $y = f(X_0')$ from some unknown $X_0'$, DPS can approximately sample from the posterior $p_\theta(X_0|y)$, utilizing the pre-trained DDPM model $p_\theta(X_0)$.

Specifically, For each DDPM step in Eq. 2, DPS includes an additional offset term that penalizes the distance between the transformed posterior mean and the measurement. More specifically, after $X_{t-1}$ is obtained from DDPM, DPS updates it additionally with:

$$X_{t-1} = X_{t-1} - \zeta_t \nabla_{X_t} \|f(\mathbb{E}[X_0|X_t]) - y\|, \quad (3)$$

where $\zeta_t$ are hyperparameters and $\mathbb{E}[X_0|X_t]$ is the posterior mean estimated by Tweedie's formula:

$$\mathbb{E}[X_0|X_t] = \frac{1}{\sqrt{\bar{\alpha}_t}}(X_t + (1 - \bar{\alpha}_t)s_\theta(X_t, t)). \quad (4)$$

In this paper, we adopt the notation of DPS in pixel space. For DPS in latent space, see Appendix. A.

## 3 DPS IS CLOSER TO MAXIMIZING A POSTERIOR

### 3.1 DPS AS CONDITIONAL SCORE ESTIMATOR

A key theoretical justification for DPS is that it can be interpreted as a conditional score estimator. Chung et al. (2022a) and Song et al. (2023c) argue that to sample from the posterior $p_\theta(X_0|y)$, one can integrate an estimate of the conditional score $\nabla_{X_t} \log p_\theta(X_t|y)$ into the DDPM update in Eq. 2 and replace the unconditional score model $s_\theta(X_t, t)$. Then, to sample from $p_\theta(X_0|y)$, we can perform ancestral sampling through a new Markov chain with the transition distribution:

$$p_\theta(X_{t-1}|X_t, y) = \mathcal{N}(\frac{1}{\sqrt{\alpha_t}}(X_t + \beta_t \nabla_{X_t} \log p_\theta(X_t|y)), \sigma_t^2 I). \tag{5}$$

To estimate the conditional score $\nabla_{X_t} \log p_\theta(X_t|y)$, Chung et al. (2022a) and Song et al. (2023c) show that it can be decomposed into the unconditional score model and a likelihood term:

$$\nabla_{X_t} \log p_\theta(X_t|y) \approx s_\theta(X_t, t) + \nabla_{X_t} \log p_\theta(y|X_t). \tag{6}$$

As $Y - X_0 - X_t$ forms a Markov chain, the term $p_\theta(y|X_t)$ can be estimated via Monte Carlo:

$$p_\theta(y|X_t) = \mathbb{E}_{p_\theta(X_0|X_t)}[p(y|X_0)] \approx \frac{1}{K} \sum_{x_0^i \sim p_\theta(X_0|X_t)}^{i=1,...,K} p(y|x_0^i), \tag{7}$$

where $p(y|X_0) \propto \exp\left(-\Delta(f(X_0), y)\right)$ and $\Delta(.,.)$ is some distance metric. When the posterior sample $x_0^i$ is approximated by the posterior mean $\mathbb{E}[X_0|X_t]$, and the distance $\Delta(.,.)$ is the $l^2$ norm weighted by $\zeta_t$, the above Monte Carlo estimation becomes DPS (Chung et al., 2022a):

$$p_\theta(y|X_t) \approx p(y|X_0 = \mathbb{E}[X_0|X_t]) = \exp\left(-\zeta_t \|f(\mathbb{E}[X_0|X_t]) - y\|\right). \tag{8}$$

Many subsequent works follow this theory to explain why DPS works (Chung et al., 2022a; Song et al., 2023c; Yu et al., 2023; Boys et al., 2023; Rout et al., 2023; Zhang et al., 2024; Chung et al., 2023; Yang et al., 2024). Chung et al. (2022a) and Song et al. (2023c) support this explanation by showing that the approximation error to $p(y|X_t)$ is upper-bounded. However, Yang et al. (2024) challenge this theory by demonstrating that the approximation error has a large lower bound.

To verify whether DPS approximates the conditional score, we first represent the conditional score estimation of DPS using the unconditional score and the $l^2$ norm:

**Proposition 1.** *DPS is equivalent to DDPM with estimated conditional score:*

$$s_{DPS}(X_t, t, y) = s_\theta(X_t, t) - \frac{\sqrt{\alpha_t}}{\beta_t} \zeta_t \nabla_{X_t} \|f(\mathbb{E}[X_0|X_t]) - y\|. \tag{9}$$

In the following sections, we will examine $s_{DPS}(X_t, t, y)$ in practical scenarios, specifically for $512 \times 512$ images, to determine whether it is a good estimation of the conditional score.

### 3.2 OBSERVATION I: DPS HAS LARGE SCORE ERROR

Our first observation is that the conditional score estimation of DPS has a large error, even when compared with the unconditional score. Furthermore, when tuning the hyperparameter $\zeta_t$, a larger score error often results in better image quality.

More specifically, we consider the scenario where $f(.)$ is $\times 8$ bicubic down-sampling. We use the score of StableSR (Wang et al., 2024), a state-of-the-art Stable Diffusion-based super-resolution method, as the baseline (See details in Appendix. B.1). We observe that starting from the score of StableSR, the distance to the score of DPS is significantly larger than the distance to the unconditional score. Denote the score of DPS as $s_{DPS}(X_t, t, y)$, the score of StableSR as $s_\theta(X_t, t, y)$ and the unconditional score as $s_\theta(X_t, t)$, we empirically observe:

$$\|s_{DPS}(X_t, t, y) - s_\theta(X_t, t, y)\| \gg \|s_\theta(X_t, t) - s_\theta(X_t, t, y)\|. \tag{10}$$

In Figure 1, we show the score error of different methods compared to StableSR. The results indicate that only when $\zeta_t = 0.05$ and $t \geq 500$ is the score error of DPS marginally smaller than that of the

unconditional score. However, when $\zeta_t = 0.05$, DPS does not work well (See Figure 2 and Table 1). For all other $\zeta_t$, the score error of DPS is significantly larger than that of the unconditional score. Additionally, Table 1 shows that as $\zeta_t$ decreases, the image quality of DPS deteriorates. This leads to an unexpected phenomenon: for DPS, a larger score error correlates with better image quality.

To verify that StableSR is a reliable baseline for conditional score, we train a super-resolution ControlNet (Zhang et al., 2023), which uses a different neural network. However, as shown in Figure 1 the score error of the ControlNet is significantly smaller than unconditional score. Besides, StableSR outperforms DPS in FID (Fréchet inception distance) (Heusel et al., 2017), KID (Kernel Inception Distance) (Binkowski et al., 2018) and LPIPS (Learned Perceptual Image Patch Similarity) (Zhang et al., 2018), confirming that it is a reliable estimate of the conditional score.

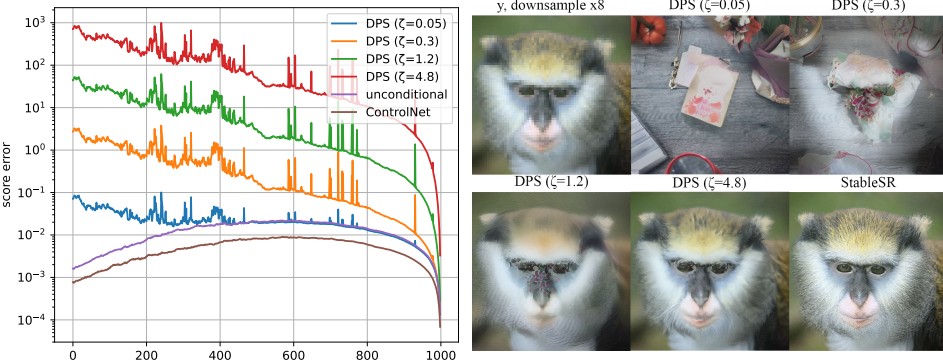

Figure 1: The score error to StableSR. DPS has a larger error than unconditional score.

Figure 2: Qualitative results of different approaches for SR×8. DPS only works with $\zeta_t = 4.8$.

Table 1: Quantitative results for SR×8. DPS only works with $\zeta_t = 4.8$.

Table 2: The mean of score for SR×8. The mean of DPS ($\zeta_t = 4.8$) is large.

| | LPIPS | FID | KID | | Score's Mean (k=1000) |
|---|---|---|---|---|---|
| DPS ($\zeta_t = 0.05$) | 0.7833 | 131.0 | 62.1e-3 | DPS ($\zeta_t = 0.05$) | 0.4058 |
| DPS ($\zeta_t = 0.3$) | 0.7349 | 150.5 | 87.6e-3 | DPS ($\zeta_t = 0.3$) | 0.5078 |
| DPS ($\zeta_t = 1.2$) | 0.6056 | 133.0 | 86.3e-3 | DPS ($\zeta_t = 1.2$) | 1.4713 |
| DPS ($\zeta_t = 4.8$) | 0.4137 | 58.48 | 14.6e-3 | DPS ($\zeta_t = 4.8$) | 5.8568 |
| ControlNet | 0.4657 | 67.38 | 19.3e-3 | Unconditional | 0.4026 |
| StableSR | 0.2855 | 29.12 | 0.9e-3 | StableSR | 0.3939 |

### 3.3 OBSERVATION II: DPS HAS LARGE SCORE MEAN

Our second observation is that the conditional score estimation of DPS has a larger non-zero mean. Additionally, when tuning the hyperparameter $\zeta_t$, we find that a larger score mean correlates with better image quality. More specifically, we empirically observe:

$$|\mathbb{E}[s_{DPS}(X_t, t, y)]| \gg |\mathbb{E}[s_\theta(X_t, t)]| \approx |\mathbb{E}[s_\theta(X_t, t, y)]| > 0 \tag{11}$$

We examine the mean of the score function as any valid score function has zero mean. This zero-mean property is widely adopted in reinforcement learning and gradient estimators (Williams, 1992; Mnih & Rezende, 2016; Mohamed et al., 2020):

$$\mathbb{E}[\nabla_X \log p(X)] = \int p(X)\nabla_X \log p(X)dX = \int \nabla_X p(X)dX = \nabla_X \int p(X)dX = 0. \tag{12}$$

In Table 2, we present the absolute value of the empirical mean of the score for different methods. Each score is computed at the initial step $X_T$ for 1000 samples. The results show that the unconditional score and StableSR have score means close to 0.4. For DPS with $\zeta_t = 0.05$ and $\zeta_t = 0.3$, the score mean is also close to 0.4, but DPS does not perform well for small $\zeta_t$. In contrast, for DPS with good practical performance ($\zeta_t = 4.8$), the score mean is close to 5.8, which is far from zero.

### 3.4 OBSERVATION III: DPS HAS LOW SAMPLE DIVERSITY

From these observations, we can conclude that DPS is not estimating the conditional score accurately. So why does DPS perform well in practice? To shed light on this, we additionally show that the sample diversity of DPS is much lower than that of well-trained conditional diffusion models such as StableSR. Denote the sample variance of DPS as $\mathbb{V}[X_0^{DPS}|y]$, and the sample variance of StableSR as $\mathbb{V}[X_0|y]$, we empirically observe:

$$\mathbb{V}[X_0^{DPS}|y] \ll \mathbb{V}[X_0|y]. \tag{13}$$

We draw $k = 50$ samples from DPS and StableSR respectively and compute the per-pixel standard deviation. The results are shown in Figure 3 and Table 3. We observe that DPS with $\zeta_t = 4.8$ has a much lower standard deviation than StableSR. In Figure 4, we show that the samples from StableSR contain more visual variations than DPS, which is also observed by Cohen et al. (2023).

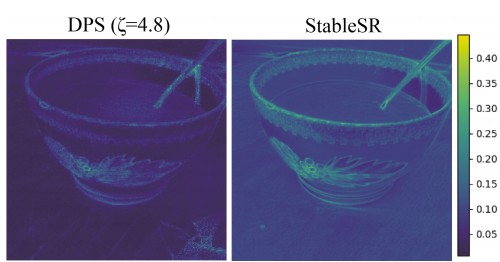

Figure 3: The per-pixel standard deviation of DPS is much lower than that of StableSR.

Table 3: The average per-pixel standard deviation of DPS is much lower than that of StableSR.

|  | Pixel's STD. (k=50) |
| --- | --- |
| DPS ($\zeta_t = 4.8$) | 0.0453 |
| StableSR | 0.3939 |

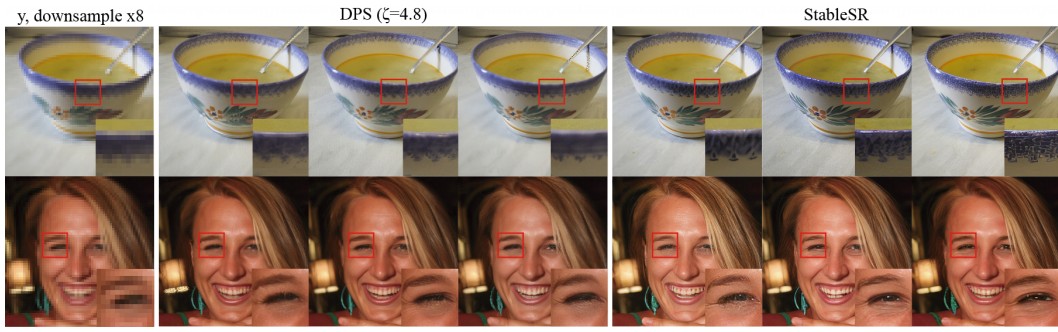

Figure 4: The image samples from DPS has much lower visual diversity than that of StableSR.

### 3.5 HYPOTHESIS: DPS ALIGNS MORE CLOSELY TO MAXIMIZING A POSTERIOR

With Observation I & II, one can conclude that DPS is not an effective conditional score estimator for image restoration. However, DPS does work well and produces high-quality samples, despite being a poor conditional score estimator. Along with Observation III that DPS has low sample diversity, we hypothesize that DPS is closer to another paradigm of image restoration other than posterior sampling: the maximization of a posterior (MAP) estimate.

**Hypothesis 2.** Instead of approximating posterior sampling

$$X_{t-1} \sim p_\theta(X_{t-1}|X_t, y), \tag{14}$$

DPS is in fact attempting to maximizing a posterior

$$X_{t-1} \leftarrow \arg\max p_\theta(X_{t-1}|X_t, y). \tag{15}$$

The MAP hypothesis is the simplest one that can explain all our observations:

- Observation I & II: Since DPS is not estimating the conditional score, its estimated score might have a large error and mean.

- Observation III: As MAP estimators produce deterministic samples, the low sample diversity of DPS is also explained.

Besides, the MAP hypothesis also explains previous works that are not explainable with conditional score estimation theory, such as why DPS has a high score error lower-bound (Yang et al., 2024) and why Adam helps DPS (He et al., 2023; Chung et al., 2023) (See Appendix C.1).

## 4 IMPROVING DPS WITH MAP HYPOTHESIS

### 4.1 IMPROVEMENT I: EXPLICIT MAP IMPLEMENTATION

In light of the MAP hypothesis, we modify DPS to explicitly maximize the posterior. In order to achieve this, we first reformulate the MAP hypothesis into a constrained optimization:

**Proposition 3.** *MAP in Eq. 15 is equivalent to the following in probability as $d \to \infty$:*

$$X_{t-1} \leftarrow \arg \max \log p_\theta(y|X_{t-1}), \text{ s.t. } X_{t-1} \sim p_\theta(X_{t-1}|X_t). \tag{16}$$

To implement the constrained optimization in Eq. 16, we leverage an observation made in Yang et al. (2024), that as dimension $d \to \infty$, the isotropic Gaussian distribution $p_\theta(X_{t-1}|X_t)$ concentrates to a sphere surface $\mathcal{S}(p_\theta(X_{t-1}|X_t))$, with radius $\sqrt{d}\sigma_t$ and center $\mathbb{E}[X_{t-1}|X_t]$. Then, we can adopt multiple steps of gradient ascent and project the result onto the sphere surface $\mathcal{S}(p_\theta(X_{t-1}|X_t))$ following Menon et al. (2020) and Yang et al. (2024). The detailed implementation is shown in Algorithm 2. It has two differences from DPS: 1). We implement the maximization of $\log p_\theta(y|X_{t-1})$ with multiple steps of gradient ascent. 2). We project the result onto the sphere surface $\mathcal{S}(p_\theta(X_{t-1}|X_t))$ where $p_\theta(X_{t-1}|X_t)$ concentrates around.

| **Algorithm 1:** DPS | **Algorithm 2:** DMAP |
|---|---|
| 1 **input** $T, f(.), y, \zeta_t$ | 1 **input** $T, K, f(.), y, \zeta_t$ |
| 2 $\quad x_T = \mathcal{N}(0, I)$ | 2 $\quad x_T = \mathcal{N}(0, I)$ |
| 3 $\quad$ **for** $t = T$ **to** 1 **do** | 3 $\quad$ **for** $t = T$ **to** 1 **do** |
| 4 $\qquad x_{t-1} \sim p_\theta(X_{t-1}|x_t)$ | 4 $\qquad x_{t-1} \sim p_\theta(X_{t-1}|x_t), \mu_{t-1} = \mathbb{E}[X_{t-1}|x_t]$ |
| 5 $\qquad x_{t-1} = x_{t-1} - \zeta_t \nabla\|f(\mathbb{E}[X_0|x_t]) - y\|$ | 5 $\qquad$ **for** $j = 1$ **to** $K$ **do** |
| 6 $\quad$ **return** $x_0$ | 6 $\qquad\qquad x_{t-1} = x_{t-1} - \zeta_t \nabla\|f(\mathbb{E}[X_0|x_{t-1}]) - y\|$ |
| | 7 $\qquad\qquad x_{t-1} = \mu_{t-1} + \sqrt{d}\sigma_t \frac{x_{t-1} - \mu_{t-1}}{\|x_{t-1} - \mu_{t-1}\|}$ |
| | 8 $\quad$ **return** $x_0$ |

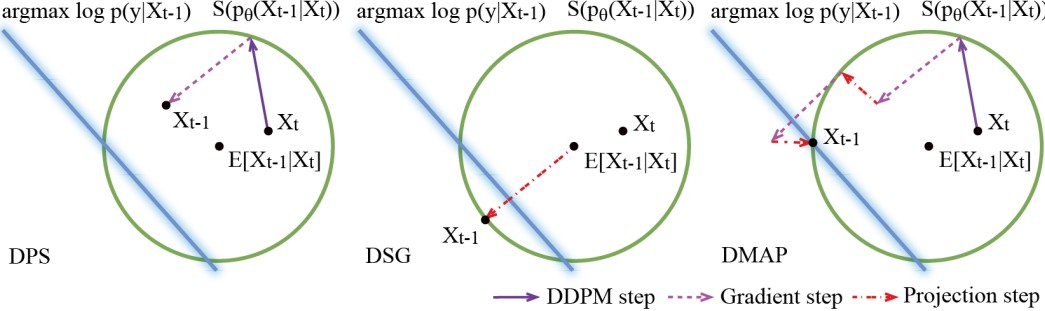

Figure 5: Conceptual illustration of the update procedure of DPS, DSG and DMAP. The green circle $\mathcal{S}(p(X_{t-1}|X_t))$ is the sphere surface that $p(X_{t-1}|X_t)$ concentrates to. The blueline $\arg \max \log p(y|X_{t-1})$ is the manifold of $X_{t-1}$ that maximizes $\log p(y|X_{t-1})$.

As Algorithm 2 is a faithful implementation of the MAP in Eq. 15, we name it Diffusion Maximize a Posterior (DMAP). To better understand why DMAP works, in Figure 5, we illustrate the update procedure of DMAP, and compare it with DPS (Chung et al., 2022a) and DSG (Yang et al., 2024). It is shown that for DPS, the resulting $X_{t-1}$ might not live on $\mathcal{S}(p_\theta(X_{t-1}|X_t))$, and might not

maximize $\log p_\theta(y|X_{t-1})$. For DSG, the resulting $X_{t-1}$ must live on $\mathcal{S}(p_\theta(X_{t-1}|X_t))$, but might not maximize $\log p_\theta(y|X_{t-1})$. While for DMAP, the resulting $X_{t-1}$ must live on $\mathcal{S}(p_\theta(X_{t-1}|X_t))$, and can better achieve maximizing $\log p_\theta(y|X_{t-1})$ due to multi-step update.

### 4.2 Improvement II: A Light-weight Conditional Score Estimator

Furthermore, the MAP hypothesis allows us to improve DPS with a reasonable but not-so-accurate conditional score estimator (CSE). Denote the backward transition distribution of CSE as $q_\theta(X_{t-1}|X_t, y)$, we can use it as the initialization point for solving the constrained optimization problem (i.e., we use an estimated $q_\theta(X_{t-1}|X_t, y)$ to replace unconditional $p_\theta(X_{t-1}|X_t)$ in Eq. 16). In other word, we iteratively optimize the following posterior using multi-step gradient ascent:

$$X_{t-1} \leftarrow \arg\max \log p_\theta(y|X_{t-1}), \text{ with init. } X_{t-1} \sim q_\theta(X_{t-1}|X_t, y). \quad (17)$$

To improve DPS, the approximate posterior $q_\theta(X_{t-1}|X_t, y)$ can be not-so-accurate. In fact, as long as the cross-entropy between the approximate posterior and the true posterior is smaller than the cross-entropy between the unconditional distribution and the true posterior, the above optimization problem has a better initialization than Eq. 16 in expectation:

**Proposition 4.** *Denote the cross entropy as* $\mathcal{H}(.,.)$, *then*

$$\mathcal{H}(q_\theta(X_{t-1}|X_t, y), p_\theta(X_{t-1}|X_t, y)) < \mathcal{H}(p_\theta(X_{t-1}|X_t), p_\theta(X_{t-1}|X_t, y))$$
$$\Rightarrow \mathbb{E}_{q_\theta(X_{t-1}|X_t, y)}[\log p_\theta(y|X_{t-1})] > \mathbb{E}_{p_\theta(X_{t-1}|X_t)}[\log p_\theta(y|X_{t-1})]. \quad (18)$$

In practice, we adopt ControlNet (Zhang et al., 2023) as CSE. We are surprised to find that our CSE is very efficient in terms of both data and temporal complexity. Specifically, it only takes 8 GPU hours and 100 images to train, which is roughly equivalent to the time of running DPS for 100 images. For DPS evaluated with 1000 images, the additional training time of CSE is around 28s for each image, while the DPS itself costs several minutes. Alternatively, we can also use fully synthetic images generated by Stable Diffusion, in which case our CSE does not require any new data.

Table 4: Quantitative Results on ImageNet 512 Dataset. **Bold**: best. Underline: second best. Our proposed approaches improve DPS significantly.

| | Time(s)↓ | SR×8 | | | Gaussian Deblur | | | Non-linear Deblur | | |
|---|---|---|---|---|---|---|---|---|---|---|
| | | PSNR↑ | LPIPS↓ | FID↓ | PSNR↑ | LPIPS↓ | FID↓ | PSNR↑ | LPIPS↓ | FID↓ |
| *Training Free* | | | | | | | | | | |
| DPS | 162 | 22.33 | 0.4137 | 58.48 | 23.05 | 0.4267 | 59.31 | 23.34 | 0.4173 | 62.10 |
| PSLD | 279 | 22.28 | 0.4163 | 59.08 | 23.06 | 0.4305 | 60.73 | - | - | - |
| FreeDOM | 195 | 22.64 | 0.3961 | 52.94 | 23.33 | 0.4104 | 54.26 | 23.39 | 0.4023 | 58.98 |
| ReSample | 433 | 22.78 | 0.3949 | 52.18 | 23.50 | 0.4076 | 53.00 | 23.47 | 0.4026 | 59.24 |
| DSG | 165 | 23.15 | 0.3912 | 51.15 | 23.70 | 0.3977 | 52.01 | 23.41 | 0.3861 | 57.81 |
| DMAP (same) | 172 | 23.37 | 0.3770 | 44.37 | 24.42 | 0.3752 | 44.15 | 24.41 | 0.3437 | 47.65 |
| DMAP (full) | 517 | **23.52** | **0.3494** | **39.56** | **24.86** | **0.3407** | **38.33** | **24.99** | **0.3135** | **39.93** |
| *Conditional Score Estimator Trained with 8 GPU Hours* | | | | | | | | | | |
| CSE (n=100) | 36 + 28 | 16.54 | 0.4841 | 85.58 | 17.54 | 0.4042 | 58.89 | 16.20 | 0.5172 | 94.68 |
| CSE (n=1000) | 35 + 28 | 16.68 | 0.4657 | 67.38 | 17.18 | 0.3946 | 49.50 | 15.69 | 0.5028 | 77.52 |
| CSE (Self-gen) | 36 + 28 | 17.15 | 0.4794 | 82.32 | 18.78 | 0.3700 | 45.21 | 16.45 | 0.5213 | 106.2 |
| *DPS + Conditional Score Estimator Trained with 8 GPU Hours* | | | | | | | | | | |
| DPS | 162 | 22.33 | 0.4137 | 58.48 | 23.05 | 0.4267 | 59.31 | 23.34 | 0.4173 | 62.10 |
| DPS + CSE (n=100) | 186 + 28 | 22.98 | 0.3229 | 44.59 | 25.20 | 0.2306 | 23.39 | 24.08 | 0.3351 | 40.25 |
| DPS + CSE (n=1000) | 184 + 28 | **23.35** | 0.3142 | **36.23** | **25.47** | **0.2150** | **18.96** | **24.17** | 0.3384 | **36.90** |
| DPS + CSE (Self-gen) | 192 + 28 | 23.09 | **0.3082** | 38.60 | 25.41 | 0.2222 | 20.88 | 24.01 | **0.3250** | 40.64 |
| *DMAP + Conditional Score Estimator Trained with 8 GPU Hours* | | | | | | | | | | |
| DMAP (full) | 517 | 23.52 | 0.3494 | 39.56 | 24.86 | 0.3407 | 38.33 | 24.99 | 0.3135 | 39.93 |
| DMAP + CSE (n=100) | 603 + 28 | 23.58 | 0.3357 | 39.56 | 26.17 | 0.2580 | 22.66 | 25.19 | 0.3048 | 33.08 |
| DMAP + CSE (n=1000) | 602 + 28 | **23.71** | 0.3254 | **36.64** | **26.39** | 0.2550 | **20.49** | **25.27** | 0.3114 | 32.97 |
| DMAP + CSE (Self-gen) | 602 + 28 | 23.33 | **0.3107** | 38.29 | 26.34 | **0.2438** | 21.17 | 25.19 | **0.2776** | **32.04** |

## 5 Experimental Results

### 5.1 Experimental Setup

**Dataset & Diffusion Model** We utilize the first 1000 images from the ImageNet validation split. All images are resized and center-cropped to $512^2$ pixels. We use Stable Diffusion 2.0 as the base diffu-

sion model and employ a 500-step ancestral sampling solver (*i.e.*, DDPM) to align with our assumptions. There are some works using ancestral sampling solvers (Yu et al., 2023; Yang et al., 2024) while some other works (Chung et al., 2023; Rout et al., 2023) using PF-ODE solvers (*i.e.*, DDIM). We discuss the impact of base diffusion models, solvers and detailed setup in Appendix. B.4.

**Operators & Metrics** For the operator $f(.)$, we adopt $\times 8$ downsampling, Gaussian blurring with a kernel size 61 and an intensity 3.0, and non-linear blurring as described by Chung et al. (2022a). Following previous works (Chung et al., 2022a), we utilize PSNR (Peak Signal-to-Noise Ratio), LPIPS, and FID to evaluate the performance of all methods.

**Baselines** We compare our two improvements against several DPS algorithms in the latent space. These include DPS (Chung et al., 2022a) implemented in latent space as described in Rout et al. (2024), PSLD (Rout et al., 2024), ReSample (Song et al., 2023a), FreeDOM (Yu et al., 2023), and DSG (Yang et al., 2024). We acknowledge the existence of other very competitive works (Song et al., 2023c; Rout et al., 2023; Chung et al., 2023; Mardani et al., 2023b; Song et al., 2023b). However, these approaches are either not open-sourced or have not been evaluated in the latent space, and hence, are not included in our comparison.

**Conditional Score Estimator** The Conditional Score Estimator (CSE) utilizes a ControlNet neural network and is trained for 5000 steps with a batch size of 64, taking approximately 8 hours on one A100 GPU. We train the CSE using various datasets, including subsets of 100 and 1000 images from the ImageNet training set, as well as self-generated data from Stable Diffusion 2.0. For detailed information about the training setup and datasets, please refer to Appendix. B.1.

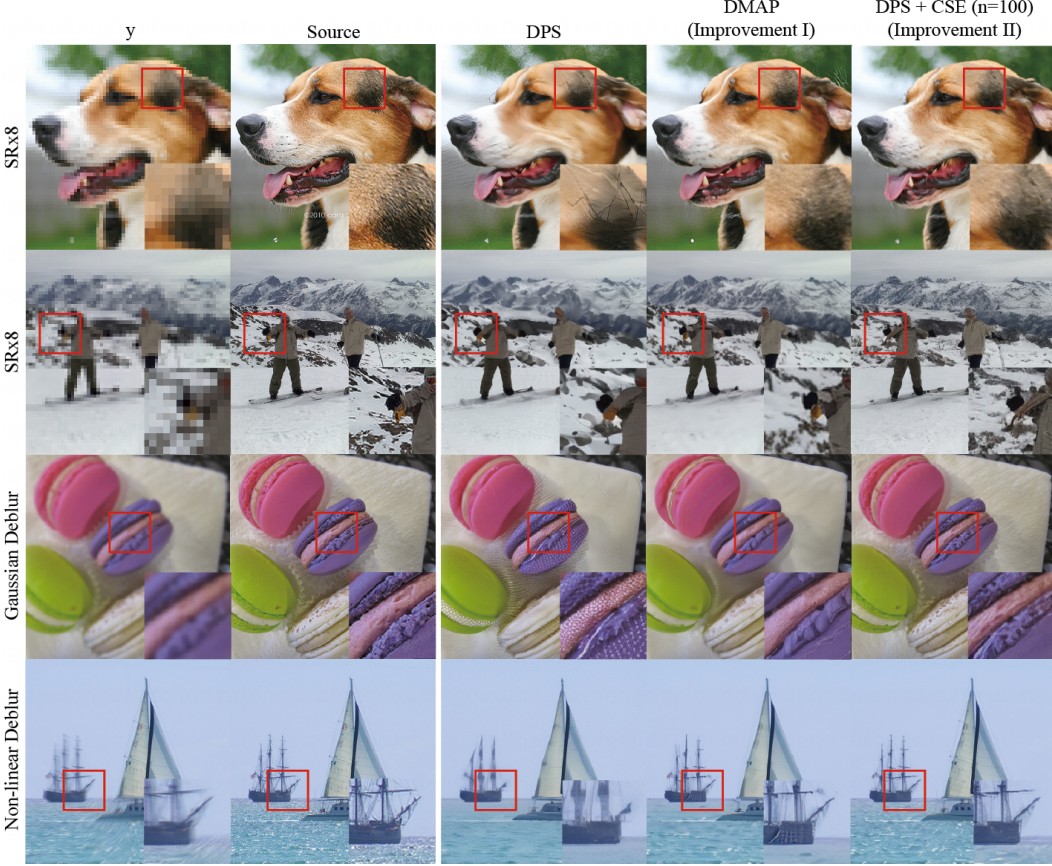

Figure 6: Qualitative results on $512 \times 512$ ImageNet images. Our proposed approaches improve DPS significantly.

## 5.2 MAIN RESULTS

**Improvement I** We evaluate DMAP in two different settings: "same complexity" and "best performance", to test DMAP as a practical algorithm and verify our MAP hypothesis. In the "same complexity" setting, we set the gradient ascent steps of DMAP to $K = 2$ and reduce the diffusion steps by 2. This results in DMAP (same), which has almost the same runtime as DPS. In the "best performance" setting, we increase the gradient ascent steps of DMAP to $K = 3$ without reducing the diffusion steps. This results in DMAP (full), which is significantly slower than DPS. As shown in Table 4 and Figure 6, 10-12, DMAP (same) outperforms all other methods across most metrics while maintaining almost identical complexity to DPS. Furthermore, DMAP (full) outperforms all other methods by a large margin, albeit at the cost of increased complexity. Therefore, we conclude that DMAP is an efficient and practical algorithm, and our MAP hypothesis is reasonable.

**Improvement II** We evaluate our Improvement II across three different dataset settings. Specifically, we train a Conditional Score Estimator (CSE) using 100 and 1000 images from ImageNet, as well as self-generated images from Stable Diffusion 2.0 itself. Each training session takes approximately 8 GPU hours. As shown in Table 4 and Figure 6, 10-12, the CSE on its own does not provide substantial results. However, when integrated with DPS and DMAP algorithms, the CSE significantly enhances their performance. Besides, the additional complexity of CSE is also marginal. More specifically, the training cost amortized by 1000 images is 28s, which is much smaller than the 162s of DPS itself. Overall, DPS + CSE is only 25% slower than DPS, including the training time.

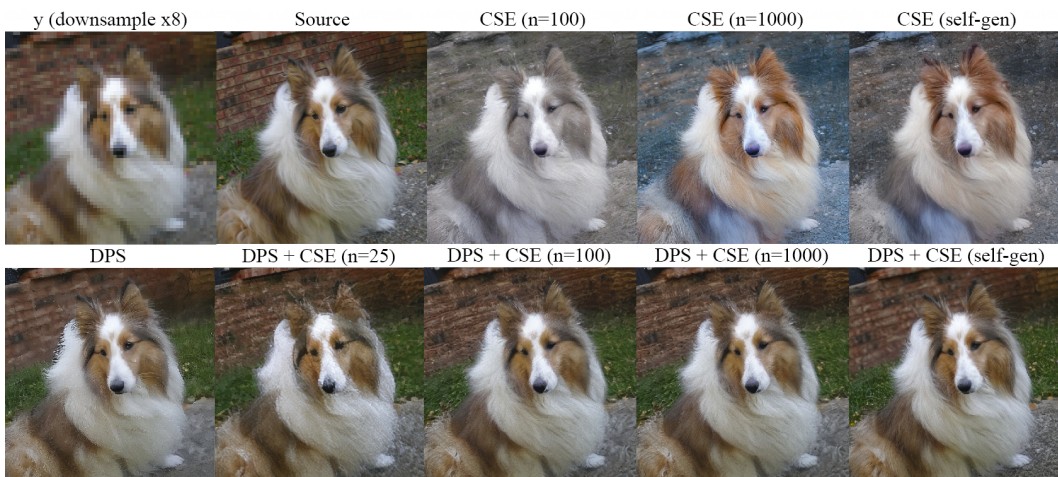

Figure 7: Effect of CSE and number of training images. CSE does not work well without DPS.

Table 5: Impact of steps parameter $T, K$.

|  | T | K | Time(s) | PSNR | LPIPS | FID |
|---|---|---|---|---|---|---|
| DPS | 100 | 1 | 42 | 20.50 | 0.5980 | 132.8 |
| DSG | 100 | 1 | 42 | 22.12 | 0.4739 | 76.24 |
| DMAP | 100 | 1 | 48 | 22.69 | 0.4763 | 80.50 |
| DMAP | 50 | 2 | 45 | 22.60 | 0.4591 | 72.05 |
| DMAP | 25 | 4 | 44 | 22.65 | 0.4681 | 76.63 |

Table 6: Impact of dataset size and type.

| # of Image | Dataset | PSNR | LPIPS | FID |
|---|---|---|---|---|
| 0 (DPS) | - | 22.33 | 0.4137 | 58.48 |
| 25 | ImageNet | 22.26 | 0.4050 | 75.64 |
| 50 | ImageNet | 22.84 | 0.3527 | 55.78 |
| 100 | ImageNet | 22.98 | 0.3229 | 44.59 |
| 1000 | ImageNet | 23.35 | 0.3142 | 36.23 |
| 100000 | ImageNet | 23.46 | 0.3147 | 35.96 |
| - | Self-gen | 23.09 | 0.3082 | 38.60 |

Table 7: Impact of GPU hours.

| # of GPU hours | PSNR | LPIPS | FID |
|---|---|---|---|
| 0 (DPS) | 22.33 | 0.4137 | 58.48 |
| 2 | 23.04 | 0.3716 | 49.18 |
| 4 | 23.15 | 0.3310 | 40.34 |
| 8 | 23.46 | 0.3147 | 35.96 |

### 5.3 ABLATION STUDIES

**Improvement I** In Table 5, we present an ablation study to examine how balancing the diffusion steps $T$ and gradient steps $K$ affects the performance of DMAP, under the constraint $T \times K = 100$. The setting of 100 diffusion steps is commonly used in fast DPS (Yang et al., 2024; He et al., 2024). Our findings indicate that for $K = 1, 2, 4$, the PSNR of DMAP does not vary significantly. However, we observe that when $K = 2$, the LPIPS and FID scores are better than those of other settings, and outperform DPS and DSG.

**Improvement II** In Table 6 and Figure 7, we present the impact of dataset size and type on our Improvement II. The results show that a CSE trained with only 100 images can significantly enhance the performance of DPS, while a CSE trained with just 25 images fails to do so, likely due to overfitting. Moreover, the benefit of increasing the dataset size becomes minimal once the dataset contains $\geq 1000$ images. Additionally, a CSE trained with self-generated images also leads to notable improvements. In Table 7, we present the impact of training time on our Improvement II. Results indicate that a CSE trained for $\geq 4$ hours can significantly enhance the performance of DPS.

## 6 RELATED WORK

Recently, zero-shot conditional sampling from diffusion models has drawn great attention. Various approaches have been developed, including linear projection (Wang et al., 2022; Kawar et al., 2022; Chung et al., 2022b; Lugmayr et al., 2022; Song et al., 2022; Dou & Song, 2023; Pokle et al., 2024; Cardoso et al., 2024), Monte Carlo sampling (Wu et al., 2024; Phillips et al., 2024; Dou & Song, 2024), and variational inference (Feng et al., 2023; Mardani et al., 2023a; Janati et al., 2024). Among these paradigms, Diffusion Posterior Sampling (DPS) and its variants are the most popular and widely adopted (Chung et al., 2022a; Song et al., 2023c; Yu et al., 2023; Boys et al., 2023; Rout et al., 2023; Zhang et al., 2024; Chung et al., 2023; Bansal et al., 2023; Yang et al., 2024). This popularity is due to the DPS family's ability to handle high-resolution images, accommodate non-linear operators, and operate with reasonable efficiency.

On the other hand, unlike Monte Carlo based approaches (Wu et al., 2024; Dou & Song, 2024), DPS is less explainable. Chung et al. (2022a) and Song et al. (2023c) propose that DPS works as a conditional score estimator, and provide an upper-bound on the estimation error. However, Yang et al. (2024) show that the estimation error has a non-trivial lower-bound for high-dimension data. To the best of our knowledge, we are the first to check the estimation error of DPS as a conditional score estimator in a real-life scenario, and the first to argue that DPS is closer to maximize a posterior (MAP). Besides the observations in this paper, our MAP assumption also explains previous observations that are contradicted to conditional score estimation assumption, such as why DPS works well despite it has large estimation error lower-bound (Yang et al., 2024), why DPS works with Adam (He et al., 2023; Chung et al., 2023), and why DPS has low sample diversity (Cohen et al., 2023).

## 7 DISCUSSION & CONCLUSION

One limitation of the conditional score estimator used in Improvement II is that it employs Control-Net (Zhang et al., 2023), which is not optimized for sample and computational efficiency. It would be particularly interesting if we could improve DPS using just a handful of examples and shorter time, extending its applicability from zero-shot to few-shot scenarios.

To conclude, we demonstrate that DPS does not function as a conditional score estimator through three key observations. Instead, we hypothesize that DPS is closer to maximizing a posterior distribution. Based on this new hypothesis, we propose two improvements to DPS. Empirical results show that both of our proposed improvements significantly enhance the performance of DPS.

### ACKNOWLEDGMENTS

This work is supported by Wuxi Research Institute of Applied Technologies, Tsinghua University under Grant 20242001120. No Patent or IP Claims.

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

# A    NOTATIONS AND PROOF OF MAIN RESULTS

## A.1    NOTATIONS

In the main text of this paper, we follow the notation of diffusion in pixel space without explicitly emphasizing the existence of the auto-encoder in latent diffusion model. Some previous works (Song et al., 2023a; Chung et al., 2023) explicitly emphasize the existence of the auto-encoder, as they address the approximation errors caused by it. However, our paper and some other works (Yang et al., 2024) omit the auto-encoder, as it is not the subject of study and is not considered very important.

For completeness, we provide the notation for latent diffusion with the auto-encoder here. More specifically, we denote the decoder as $\mathcal{D}(.)$ and the diffusion state in latent space as $Z_t$. Consequently, the DPS update can be expressed as:

$$\text{Pixel diffusion notation: } X_{t-1} = X_{t-1} - \zeta_t \nabla_{X_t} \|f(\mathbb{E}[X_0|X_t]) - y\|, \tag{19}$$
$$\text{Latent diffusion notation: } Z_{t-1} = Z_{t-1} - \zeta_t \nabla_{Z_t} \|f(\mathcal{D}(\mathbb{E}[Z_0|Z_t])) - y\|.$$

All other formulas can be seamlessly transferred into the latent space by replacing $\|f(\mathbb{E}[X_0|X_t]) - y\|$ with $\|f(\mathcal{D}(\mathbb{E}[Z_0|Z_t])) - y\|$. For instance, we provide the algorithms for DPS and DMAP in the latent space as shown in Algorithm 3 and Algorithm 4, respectively.

| **Algorithm 3:** Latent DPS | **Algorithm 4:** Latent DMAP |
|---|---|
| 1 **input** $T, f(.), y, \zeta_t$ | 1 **input** $T, K, f(.), y, \zeta_t$ |
| 2    $z_T = \mathcal{N}(0, I)$ | 2    $z_T = \mathcal{N}(0, I)$ |
| 3    **for** $t = T$ **to** 1 **do** | 3    **for** $t = T$ **to** 1 **do** |
| 4        $z_{t-1} \sim p_\theta(Z_{t-1}|z_t)$ | 4        $z_{t-1} \sim p_\theta(Z_{t-1}|z_t), \mu_{t-1} = \mathbb{E}[Z_{t-1}|z_t]$ |
| 5        $z_{t-1} = x_{t-1} - \zeta_t \nabla \|f(\mathcal{D}(\mathbb{E}[Z_0|z_t])) - y\|$ | 5        **for** $j = 1$ **to** $K$ **do** |
| 6    **return** $\mathcal{D}(z_0)$ | 6            $z_{t-1} = z_{t-1} - \zeta_t \nabla \|f(\mathcal{D}(\mathbb{E}[Z_0|z_{t-1}])) - y\|$ |
| | 7            $z_{t-1} = \mu_{t-1} + \sqrt{d}\sigma_t \frac{z_{t-1}-\mu_{t-1}}{\|z_{t-1}-\mu_{t-1}\|}$ |
| | 8    **return** $\mathcal{D}(z_0)$ |

## A.2    PROOF OF MAIN RESULTS

**Proposition 1.** *DPS is equivalent to DDPM with estimated conditional score:*

$$s_{DPS}(X_t, t, y) = s_\theta(X_t, t) - \frac{\sqrt{\alpha_t}}{\beta_t} \zeta_t \nabla_{X_t} \|f(\mathbb{E}[X_0|X_t]) - y\|. \tag{9}$$

*Proof.* Take Eq. 9 into the DDPM update in Eq. 2, we have

$$p_\theta(X_{t-1}|X_t, y) = \mathcal{N}(\frac{1}{\sqrt{\alpha_t}}(X_t + \beta_t s_{DPS}(X_t, t, y)), \sigma_t^2 I)$$

$$= \mathcal{N}(\frac{1}{\sqrt{\alpha_t}}(X_t + \beta_t s_\theta(X_t, t)), \sigma_t^2 I) - \zeta_t \nabla_{X_t} \|f(\mathbb{E}[X_0|X_t]) - y\|, \tag{20}$$

which is equivalent to the DDPM sampling followed by a DPS update. $\square$

**Proposition 3.** *MAP in Eq. 15 is equivalent to the following in probability as $d \to \infty$:*

$$X_{t-1} \leftarrow \arg\max \log p_\theta(y|X_{t-1}), \text{ s.t. } X_{t-1} \sim p_\theta(X_{t-1}|X_t). \tag{16}$$

*Proof.* We can rewrite the optimization target in Eq. 15 as

$$\arg\max \log p_\theta(X_{t-1}|X_t, y) \overset{(a)}{=} \arg\max \log p_\theta(X_{t-1}, y|X_t)/p_\theta(y|X_t)$$

$$\overset{(b)}{=} \arg\max \log p_\theta(X_{t-1}, y|X_t) \tag{21}$$

$$\overset{(c)}{=} \arg\max \log p_\theta(y|X_{t-1}, X_t)p_\theta(X_{t-1}|X_t), \tag{22}$$

where (b) holds since $p_\theta(y|X_t)$ is fixed and (a)(c) hold due to the Bayesian rule. As $y - X_0 - ... - X_{t-1} - X_t$ forms a Markov Chain, we have

$$p_\theta(y|X_{t-1}, X_t) = p_\theta(y|X_{t-1}). \tag{23}$$

Then we have

$$\arg\max \log p_\theta(X_{t-1}|X_t, y) = \arg\max \log p_\theta(y|X_{t-1}, X_t) p_\theta(X_{t-1}|X_t)$$
$$= \arg\max \log p_\theta(y|X_{t-1}) p_\theta(X_{t-1}|X_t). \tag{24}$$

Following Theorem 3.1.1 of Cover (1999), we know that as the dimension $d \to \infty$, the log likelihood of the isotropic Gaussian distribution $p(X_{t-1}|X_t)$ converge to a constant in probability:

$$\log p_\theta(X_{t-1}|X_t) \xrightarrow{P} -\frac{d}{2}\ln(2\pi e\sigma^2). \tag{25}$$

Therefore $\forall x_{t-1} \sim p_\theta(X_{t-1}|X_t)$, the likelihood is a constant in the limit. Then, the optimization target can be rewritten into

$$\arg\max p_\theta(y|X_{t-1}) \text{ s.t.} X_{t-1} \sim p_\theta(X_{t-1}|X_t). \tag{26}$$

This completes the proof. $\qquad\square$

**Proposition 4.** *Denote the cross entropy as $\mathcal{H}(.,.)$, then*

$$\mathcal{H}(q_\theta(X_{t-1}|X_t, y), p_\theta(X_{t-1}|X_t, y)) < \mathcal{H}(p_\theta(X_{t-1}|X_t), p_\theta(X_{t-1}|X_t, y))$$
$$\Rightarrow \mathbb{E}_{q_\theta(X_{t-1}|X_t, y)}[\log p_\theta(y|X_{t-1})] > \mathbb{E}_{p_\theta(X_{t-1}|X_t)}[\log p_\theta(y|X_{t-1})]. \tag{18}$$

*Proof.* As $y - X_{t-1} - X_t$ forms a Markov chain, we have

$$p_\theta(X_{t-1}|X_t, y) = p_\theta(y|X_{t-1})p_\theta(X_{t-1}|X_t)/p_\theta(y|X_t). \tag{27}$$

Then we can rewrite cross entropy as

$$\mathcal{H}(q_\theta(X_{t-1}|X_t, y), p_\theta(X_{t-1}|X_t, y)) < \mathcal{H}(p_\theta(X_{t-1}|X_t), p_\theta(X_{t-1}|X_t, y))$$
$$\Rightarrow \mathbb{E}_{q_\theta(X_{t-1}|X_t, y)}[\log p_\theta(y|X_{t-1}) + \log p_\theta(X_{t-1}|X_t)] > \mathbb{E}_{p_\theta(X_{t-1}|X_t)}[\log p_\theta(y|X_{t-1}) + \log p_\theta(X_{t-1}|X_t)] \tag{28}$$

As cross entropy is always larger than entropy (Cover, 1999), for any distribution $q \neq p(X_{t-1}|X_t)$, we have

$$\mathbb{E}_q[-\log p_\theta(X_{t-1}|X_t)] > \mathbb{E}_{p_\theta(X_{t-1}|X_t)}[-\log p_\theta(X_{t-1}|X_t)]$$
$$\Rightarrow \mathbb{E}_q[\log p_\theta(X_{t-1}|X_t)] < \mathbb{E}_{p_\theta(X_{t-1}|X_t)}[\log p_\theta(X_{t-1}|X_t)]. \tag{29}$$

Taking Eq. 29 into Eq. 28, we have our result

$$\mathbb{E}_{q_\theta(X_{t-1}|X_t, y)}[\log p_\theta(y|X_{t-1})] > \mathbb{E}_{p_\theta(X_{t-1}|X_t)}[\log p_\theta(y|X_{t-1})]. \tag{30}$$

$\square$

## A.3 ADDITIONAL THEORETICAL RESULTS

For now, we can not derive MAP directly from DPS. However, with additional assumptions, it is possible to derive MAP directly from a variant of DPS: DSG (Yang et al., 2024). More specifically, DSG improves DPS with a spherical projection. The update of DSG is as follows:

$$X_{t-1} = \mathbb{E}[X_{t-1}|X_t] - \sqrt{n}\sigma_t \frac{\nabla_{X_t}||f(\mathbb{E}[X_0|X_t]) - y||}{||\nabla_{X_t}||f(\mathbb{E}[X_0|X_t]) - y||||}, \tag{31}$$

**Assumption 5.** We need some additional assumptions:

1. The posterior mean $\mathbb{E}[X_0|X_t]$ is a perfect approximation to posterior samples from $p(X_0|X_t)$. In that case, the approximation error in Eq. 8 diminishes, *i.e.*,:

$$p_\theta(y|X_t) = p(y|X_0 = \mathbb{E}[X_0|X_t]) = \exp{-\zeta_t||f(\mathbb{E}[X_0|X_t]) - y||}. \tag{32}$$

2. The $\sigma_t^2$ in Eq. 5 is so small that $\log p(y|X_{t-1})$ is locally linear in range of $\sqrt{n}\sigma_t$. The $X_t, X_{t-1}$ is so close that $\nabla_{X_{t-1}} \log p(y|X_{t-1}) \approx \nabla_{X_t} \log p(y|X_t)$.

Under those assumptions, we can show that DSG is also a MAP:

**Proposition 6.** *The DSG solves the MAP problem $X_{t-1} \leftarrow \arg\max p_\theta(X_{t-1}|X_t, y)$ in Hypothesis 2 when dimension $n$ is high.*

*Proof.* We first consider a gradient ascent with learning rate $\alpha_t$:

$$X_{t-1} = \mathbb{E}[X_{t-1}|X_t] + \alpha_t \nabla_{X_t} \log p(y|X_t).$$

As we have assumed in Assumption 5.2, the $\log p(y|X_t)$ is locally linear and $\nabla_{X_{t-1}} \log p(y|X_{t-1}) \approx \nabla_{X_t} \log p(y|X_t)$. As gradient is the direct of steepest descent, when $\alpha_t$ changes, the $X_{t-1} = \mathbb{E}[X_{t-1}|X_t] + \alpha_t \nabla_{X_t} \log p(y|X_t)$ is the maximizer of $\log p(y|X_{t-1})$, given the distance from the point $X_{t-1}$ to center $\mathbb{E}[X_{t-1}|X_t]$ is fixed. On the other hand, the DSG can be written as the above gradient ascent

$$X_{t-1} = \mathbb{E}[X_{t-1}|X_t] - \sqrt{n}\sigma_t \frac{\nabla_{X_t}||f(\mathbb{E}[X_0|X_t]) - y||}{||\nabla_{X_t}||f(\mathbb{E}[X_0|X_t]) - y||||} \tag{33}$$

$$\overset{a}{=} \mathbb{E}[X_{t-1}|X_t] + \underbrace{\frac{\sqrt{n}\sigma_t}{||\nabla_{X_t} f(\mathbb{E}[X_0|X_t]) - y||||}}_{\alpha_t} \nabla_{X_t} \log p(y|X_t). \tag{34}$$

where (a) holds due to Assumption 5.1. We further notice that DSG is equivalent to the above gradient ascent. Further, from the update procedure of DSG, the distance of $X_{t-1}$ to $\mathbb{E}[X_0|X_t]$ is $\sqrt{n}\sigma_t$. Therefore, the result of DSG is the minimizer of $\nabla_{X_{t-1}} p(y|X_{t-1})$, on the sphere surface centred at $\mathbb{E}[X_{t-1}|X_t]$ with radius $\sqrt{n}\sigma_t$:

$$X_{t-1} \leftarrow \arg\max \log p(y|X_{t-1}), X_{t-1} \in \mathcal{S}(\mathbb{E}[X_{t-1}|X_t], \sqrt{n}\sigma_t),$$

where $\mathcal{S}(\mathbb{E}[X_{t-1}|X_t], \sqrt{n}\sigma_t)$ is the sphere surface centred at $\mathbb{E}[X_{t-1}|X_t]$ with radius $\sqrt{n}\sigma_t$. Besides, from (Yang et al., 2024), we know that as dimension $n \to \infty$, $X_{t-1} \sim p(X_{t-1}|X_t)$ is equivalent to $X_{t-1} \in \mathcal{S}(\mathbb{E}[X_{t-1}|X_t], \sqrt{n}\sigma_t)$. Then as dimension $n \to \infty$, DSG is equivalent to:

$$X_{t-1} \leftarrow \arg\max \log p(y|X_{t-1}), X_{t-1} \, p(X_{t-1}|X_t),$$

which is again equivalent to Hypothesis 2 by Proposition 3. $\square$

Despite DSG being already MAP under Assumption 5, our method remains to be MAP without it. Therefore, DSG alone can not replace DMAP, and its performance is inferior to DMAP as shown in Table 4.

With Assumption 5.1, the original DPS is equivalent to classifier-based guidance (Dhariwal & Nichol, 2021). What is classifier-based guidance on earth theoretically remains an open problem. For now, we leave the strict theoretical relationship between original DPS and MAP to future works. Once this problem is solved, the nature of classifier-based guidance can be revealed. We believe this problem is beyond the scope of a paper focused on DPS.

We can think DPS as a simplification of DMAP, with one step gradient ascent and without spherical projection. As long as the step size of diffusion model is small, the $\log p(y|X_{t-1})$ can be locally linear (Assumption 5.2). In that case, one step of gradient ascent is enough. As long as the DPS guidance step is much smaller than the DDPM step, the distance from $X_{t-1}$ to the sphere surface $\mathcal{S}(\mathbb{E}[X_{t-1}|X_t], \sqrt{n}\sigma_t)$ is small. In that case, spherical projection is no longer necessary. And when those conditions are approximately satisfied, DPS also works well.

### A.4 A TOY EXAMPLE

To better understand the relationship between DPS and MAP, we provide a toy example:

- The source distribution $p(X_0)$ is 2D 2-GMM (Gaussian Mixture Model). The centres are $(-1, -1), (+1, +1)$ and the diagonal $\sigma_0 = 0.3$.

- The operator f is inpainting. More specifically, the second dimension of the random variable is set to 0, the measurement $y = (0.5, 0)$.
- The diffusion is 100 steps Karras diffusion (Karras et al., 2022) with $\sigma_{max} = 4$ and $\rho = 7$.
- The diffusion scheduler is Euler Ancestral, and the DPS $\zeta_t = 0.05$.

In Figure 8, it is shown that the true posterior is a two mode distribution. However, the samples of DPS cover only one mode of true posterior. In that example, the behaviour of DPS is closer to MAP.

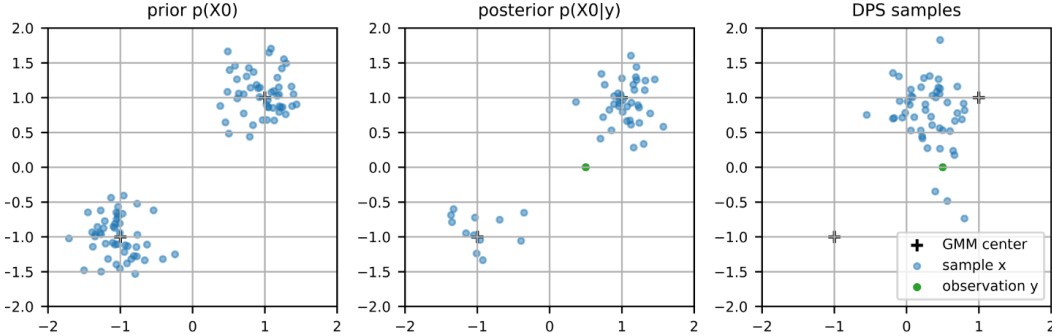

Figure 8: A toy example with 2D 2-GMM and inpainting operator. The true posterior is a 2 modal distribution, while the samples of DPS concentrate to one mode.

## B  ADDITIONAL EXPERIMENTS

### B.1  ADDITIONAL EXPERIMENTAL SETUP

All experiments were conducted on a computer equipped with an Nvidia A100 GPU, CUDA 12.1, and PyTorch 2.0. For the base diffusion model, we used the official Stable Diffusion 2.0 base model. The checkpoint for this model can be found at `https://huggingface.co/stabilityai/stable-diffusion-2`.

The score error plot in Figure. 1 is computed using the image Figure. 2, which is `ILSVRC2012_val_00000013.png` of ImageNet dataset. Similarly, the pixel variance in Table. 3 using the image in Figure. 3, which is `ILSVRC2012_val_00000004.png` of ImageNet dataset.

### B.2  ADDITIONAL DETAILS ON CSE

For the conditional score estimator (CSE), we followed the diffusers' implementation of ControlNet (Zhang et al., 2023). This implementation is available at `https://github.com/huggingface/diffusers/blob/main/examples/controlnet/train_controlnet.py`. Specifically, we used an Adam (Kingma & Ba, 2014) optimizer with a batch size of 64 and a learning rate of $5 \times 10^{-6}$. We trained the ControlNet with an $\epsilon$-prediction target for 5000 steps, which took approximately 8 hours on a single Nvidia A100 GPU. Data augmentation techniques used include random resizing with a scale range of 0.8 to 2.0, random horizontal flipping with a probability of 0.5, and random color jittering with a strength of 0.1.

### B.3  ADDITIONAL DETAILS ON BASELINES AND HYPER-PARAMETERS

**PSLD** (Rout et al., 2024) includes an additional gluing term to the original DPS for better performance in the latent space. Specifically, denoting the encoder of latent diffusion as $\mathcal{E}(.)$ and the decoder as $\mathcal{D}(.)$, the PSLD update is given by:

$$Z_{t-1} = Z_{t-1} - \zeta_t \nabla_{Z_t} \| f(\mathbb{E}[Z_0|Z_t]) - y \|, \tag{35}$$
$$-\gamma_t \nabla_{Z_t} \| \mathbb{E}[Z_0|Z_t] - \mathcal{E}(f^T(y) + \mathcal{D}(\mathbb{E}[Z_0|Z_t])) - f^T(f(\mathcal{D}(\mathbb{E}[Z_0|Z_t]))))) \|,$$

Table 8: Detailed hyper-parameters for different methods and operators.

| | SR $\times 8$ | Gaussian Deblur, Non-linear Deblur |
|---|---|---|
| DPS | $T = 500, \zeta_t = 4.8$ | $T = 500, \zeta_t = 0.6$ |
| PSLD | $T = 500, \zeta_t = 4.8, \gamma_t = 0.1$ | $T = 500, \zeta_t = 0.6, \gamma_t = 0.1$ |
| FreeDOM | $T = 500, \zeta_t = 4.8, K = 2, [c_1, c_2] = 100, 250$ | $T = 500, \zeta_t = 0.6, K = 2, [c_1, c_2] = 100, 250$ |
| ReSample | $T = 500, \zeta_t = 4.8, K = 200, N = 50, [c_1, c_2] = 100, 250$ | $T = 500, \zeta_t = 0.6, K = 200, N = 50, [c_1, c_2] = 100, 250$ |
| DSG | $T = 500, \zeta_t = 0.08$ | $T = 500, \zeta_t = 0.02$ |
| DMAP (same) | $T = 250, K = 2, \zeta_t = 9.6$ | $T = 250, K = 2, \zeta_t = 1.2$ |
| DMAP (full) | $T = 500, K = 3, \zeta_t = 4.8$ | $T = 500, K = 3, \zeta_t = 0.6$ |
| DPS + CSE | | Same as DPS |
| DMAP + CSE | | Same as DMAP |

where $f^T(.)$ is the transpose of the operator $f(.)$. Therefore, PSLD is applicable only to linear operators. In Table 8, we present how the parameters $\zeta_t$ and $\gamma_t$ are chosen for different tasks.

**FreeDOM** (Yu et al., 2023) enhances DPS by incorporating an additional step where the algorithm moves back and forth along the backward Markov chain. This approach is termed *efficient time-travel*. Specifically, for $t \in [c_1, c_2]$, FreeDOM moves back to time $t + 1$ for $K$ iterations by adding noise back.

$$X_t = \sqrt{1 - \beta_t} X_{t-1} + \mathcal{N}(0, \beta_t I). \tag{36}$$

In this way, FreeDOM effectively performs $K$ steps of gradient ascent. Given a small range for $[c_1, c_2]$, FreeDOM does not significantly increase computational complexity. In Table 8, we present how the parameters $\zeta_t$, $K$, $c_1$, and $c_2$ are chosen for different tasks.

**ReSample** (Song et al., 2023a) builds upon the *efficient time-travel* framework of FreeDOM. In addition, it further optimizes the image or latent representation directly and projects them back by adding noise. Specifically, for every $N$ steps, ReSample solves the optimization problem by performing $K$ steps of gradient ascent in the pixel space or latent space:

$$\text{Pixel space optimization: } Z^* = \mathcal{E}(\arg \min_X \frac{1}{2} \|y - f(X)\|), \tag{37}$$

$$\text{Latent space optimization: } Z^* = \arg \min_Z \frac{1}{2} \|y - f(\mathcal{D}(Z))\|,$$

with initialization by Tweedie's formula:

$$\text{Pixel space initialization: } X = \mathcal{D}(\mathbb{E}[X_0|X_t]), \tag{38}$$

$$\text{Latent space initialization: } Z = \mathbb{E}[X_0|X_t].$$

After this update, ReSample projects $Z^*$ back by adding noise back with hyper-parameter $\eta_t$:

$$Z_{t-1} = \frac{\eta_t \sqrt{\bar{\alpha}_t} \mathbb{E}[Z_0|Z_t] + (1 - \bar{\alpha}_t) Z^*}{\eta_t + (1 - \bar{\alpha}_t)} + \mathcal{N}(0, \frac{\eta_t(1 - \bar{\alpha}_t)}{\eta_t + (1 - \bar{\alpha}_t)} I). \tag{39}$$

In practice, ReSample does not significantly increase the complexity of DPS, thanks to the skipping parameter $N$. In Table 8, we present how the parameters $\zeta_t$, $K$, and $N$ are chosen for different tasks.

**DSG** (Yang et al., 2024) identifies that the distribution $p(X_{t-1}|X_t)$ concentrates on the surface of a sphere. Therefore, DSG proposes to project the result of DPS onto this sphere. DSG does not perform the DPS update directly. Instead, it computes the scaled norm of the update:

$$u^* = -\sqrt{d}\sigma_t \frac{\nabla_{X_t} \|f(\mathbb{E}[X_0|X_t]) - y\|}{\|\nabla_{X_t}\|f(\mathbb{E}[X_0|X_t]) - y\|\|}, \tag{40}$$

so that the norm of $u^*$ matches the norm of the DDPM noise $\mathcal{N}(0, \sigma_t^2 I)$. Then, DSG performs the update by controlling the strength parameter $\zeta_t$:

$$\epsilon = \mathcal{N}(0, \sigma_t^2 I), \tag{41}$$

$$u^m = \epsilon + \zeta_t(u^* - \epsilon),$$

$$X_{t-1} = \mathbb{E}[X_{t-1}|X_t] + \sqrt{d}\sigma_t \frac{u^m}{\|u^m\|}.$$

DSG does not increase the complexity of DPS. In Table 8, we present how the parameters $\zeta_t$ are chosen for different tasks.

**DMAP** For DMAP with similar complexity to DPS, we set $K = 2$ and reduce the number of diffusion steps $T$ by half, ensuring that DMAP (same) has comparable complexity to DPS. Consequently, we double $\zeta_t$ to compensate for the reduced number of diffusion steps. For DMAP with full performance, we set $K = 3$ while keeping other parameters the same as in DPS.

**Methods with CSE** For methods that are used with the Conditional Score Estimator (CSE), we keep the parameters the same as those used in the methods without CSE.

### B.4    ABLATION STUDY ON DIFFUSION MODELS AND SOLVERS

Table 9: Ablation study on base diffusion model and solvers.

| Diffusion | Solver | PSNR | LPIPS | FID |
|---|---|---|---|---|
| Stable Diffusion 2.0 | Ancestral Sampling (DDPM) | 22.33 | 0.4137 | 58.48 |
| Stable Diffusion 2.0 | PF-ODE (DDIM) | 22.08 | 0.4229 | 63.82 |
| Stable Diffusion 1.5 | Ancestral Sampling (DDPM) | 22.59 | 0.4074 | 69.74 |

**Stable Diffusion 1.5 vs. Stable Diffusion 2.0** Several previous works (Song et al., 2023a; Rout et al., 2024) have adopted Stable Diffusion 1.5 as the base model. Other works have not specified the version of Stable Diffusion used (Chung et al., 2023; Rout et al., 2023). To better understand the effect of the Stable Diffusion version, we compare the performance of Stable Diffusion 1.5 and Stable Diffusion 2.0. In our comparison, the operator is downsampling $\times 8$, the hyper-parameters are $T = 500$ and $\zeta_t = 4.8$, and the algorithm chosen is simply DPS.

| y (downsample x8) | Source | DPS with SD1.5 | DPS with SD2.0 |
|---|---|---|---|

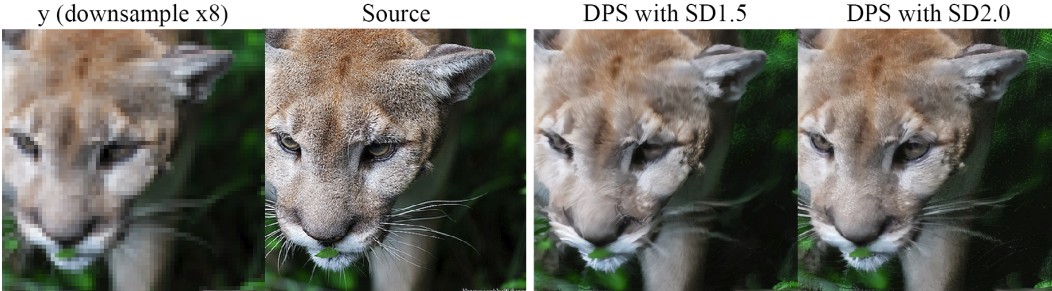

Figure 9: Visual comparison of DPS reconstruction with Stable Diffusion 1.5 and 2.0.

In Table 9, it is shown that adopting Stable Diffusion 2.0 improves the performance of DPS significantly in terms of FID, while causing slight performance degradation in terms of PSNR and LPIPS. Furthermore, as shown in Figure 9 Stable Diffusion 2.0 looks much better visually. We have not included Stable Diffusion 3.0 in the comparison because it is a flow-matching model (Lipman et al., 2022) rather than VP/VE diffusion. For such models, flow-matching specific inversion methods, such as those proposed by Pokle et al. (2023), might be more appropriate.

**Ancestral Sampling vs. PF-ODE** The original DPS (Chung et al., 2022a) for pixel diffusion adopts ancestral sampling solvers such as DDPM. Some later works on latent diffusion adopt Probability Flow Ordinary Differential Equation (PF-ODE) solvers, such as DDIM (Song et al., 2023a; Chung et al., 2023; Rout et al., 2023), while other works on latent diffusion continue to use ancestral sampling solvers (Yu et al., 2023; Yang et al., 2024). To better understand the effect of solvers, we compared the performance of ancestral sampling and PF-ODE solvers. For this comparison, the base diffusion model is Stable Diffusion 2.0, the operator is downsampling by a factor of 8, the hyper-parameters are $T = 500$ and $\zeta_t = 4.8$, and the chosen algorithm is simply DPS.

In Table 9, it is shown that adopting ancestral sampling marginally improves the performance of DPS. This observation aligns with previous findings that ancestral sampling outperforms PF-ODE when the number of diffusion steps is relatively large (Deveney et al., 2023).

## B.5 Additional Qualitative Results

In Figure 10-12, we present additional qualitative results for different methods.

## B.6 Additional Quantitative Results

In additional to super-resolution and deblurring, we present experimental results for another two operators: image inpainting and segmentation. For inpainting, we choose a fix box of size $128 \times 128$ centered at the image. For segmentation, we choose bedroom segmentation model by Lin et al. (2018). For inpainting, we adopt first 100 images of ImageNet. For segmentation, we adopt first 100 images of LSUN bedroom. For segmentation, image restoration metrics are no longer meaningful. Therefore, we use another set of metrics, including MIoU (Mean Intersection over Union) and FID. In Table 10, we show that for inpainting and segmentation, our proposed approaches (DMAP and DPS+CSE) also improve DPS significantly.

Table 10: Additional quantitative results for inpainting and segmentation.

|  | Time(s)↓ | Inpainting | | | Bedroom Segmentation | |
|---|---|---|---|---|---|---|
|  |  | PSNR↑ | LPIPS↓ | FID↓ | MIoU↑ | FID↓ |
| DPS | 173 | 23.72 | 0.3189 | 115.4 | 0.3358 | 116.7 |
| PSLD | 230 | 23.92 | 0.3279 | 117.6 | - | - |
| DSG | 193 | 23.73 | 0.3395 | 112.9 | 0.3790 | 114.7 |
| DMAP (same) | 187 | 24.50 | 0.2958 | 111.8 | 0.3883 | 102.7 |
| DMAP (full) | 484 | 24.68 | 0.2828 | 106.3 | 0.4188 | 104.9 |
| DPS + CSE (n=100) | 189 + 28 | 23.98 | 0.2618 | 74.7 | 0.3832 | 110.3 |
| DPS + CSE (Self-gen) | 189 + 28 | 24.03 | 0.2523 | 76.4 | 0.4159 | 107.8 |

# C Additional Discussions

## C.1 Additional Explanations to Previous Works

Besides the three observations, our MAP hypothesis can also explain previous works that are hard to explain using conditional score estimation assumption:

- He et al. (2023) and Chung et al. (2023) observe that Adam (Kingma & Ba, 2014) helps DPS, which is not reasonable if DPS is a conditional score estimator, because Adam is an adaptive learning rate optimizer. In other words, Adam scales the score $\nabla_{X_t} \log p(y|X_t)$ unevenly in different dimensions. This scaling, theoretically, makes the estimation of $\nabla_{X_t} \log p(y|X_t)$ inaccurate, and should deteriorate the performance of DPS. However, if we assume DPS is solving the MAP in Eq. 16, then Adam, as a general method of gradient ascent, may not harm the performance of DPS.

- Yang et al. (2024) prove that the conditional score estimation of DPS has a large approximation lowerbound, which increase as data dimension increase. If we follow the conditional score estimation assumption, DPS will not work at all for high dimension data such as images.

- Cohen et al. (2023) observe that the reconstruction of DPS lacks perceptually meaningful variants, and propose a greedy sampling technique to improve DPS's sample diversity. This visual observation is quantified in our Observation III.

ETHICS STATEMENT

The approach proposed in this paper allows for conditional generation without the need to train a new model. This not only saves the energy that would typically be used to train a conditional generative diffusion model but also decreases the related carbon emissions. Nevertheless, the possible negative effects are similar to those of other conditional generative models, including issues related to trust and the ethical considerations of producing fake images.

REPRODUCIBILITY STATEMENT

For theoretical results, the proofs for all theorems are presented in Appendix A. For the experiments, we used four publicly accessible datasets. Additional implementation details are provided in Appendix B.1. Moreover, we include the source code for reproducing the experimental results in the supplementary material.

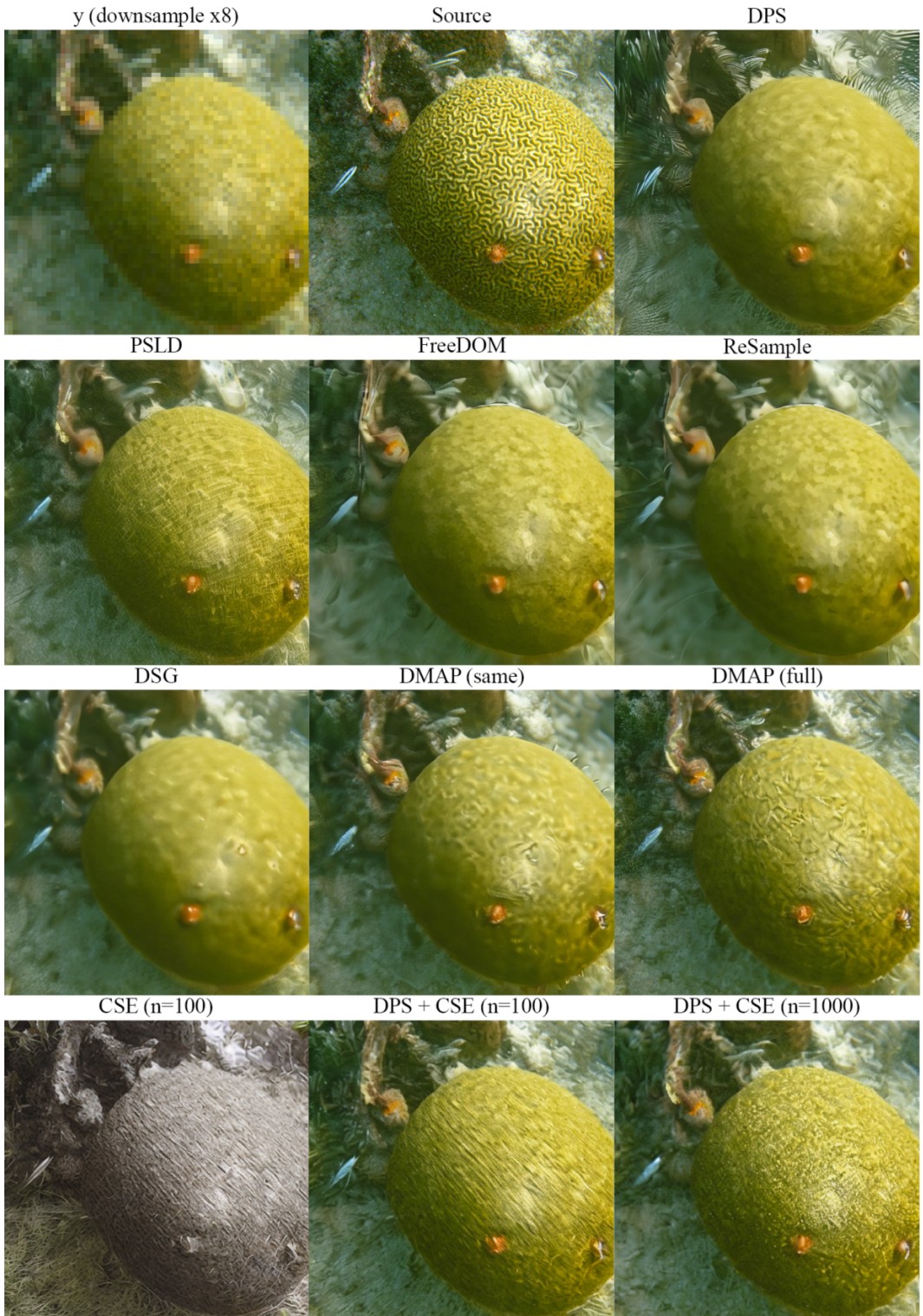

Figure 10: Additional qualitative results for SR×8.

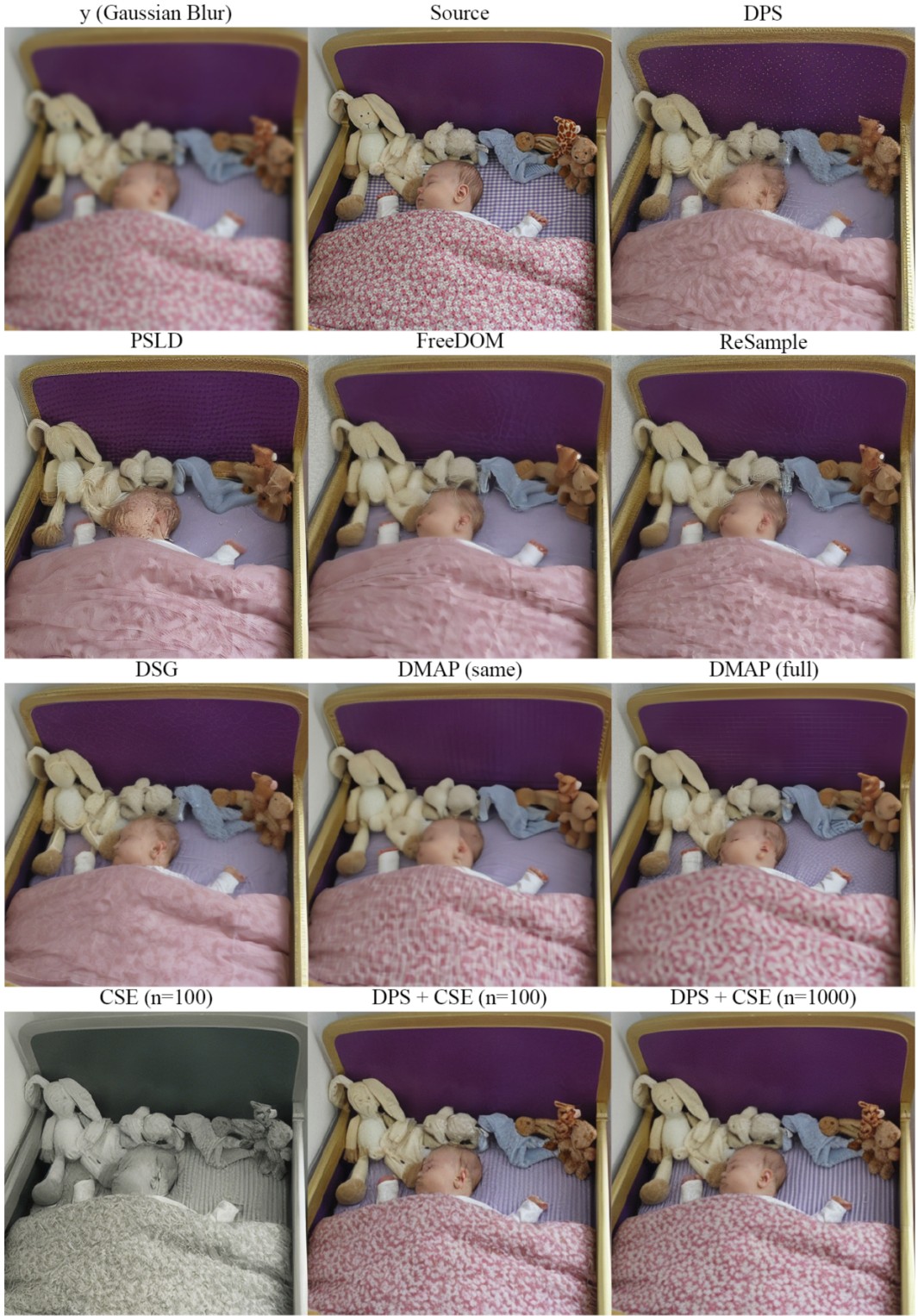

Figure 11: Additional qualitative results for Gaussian deblurring.

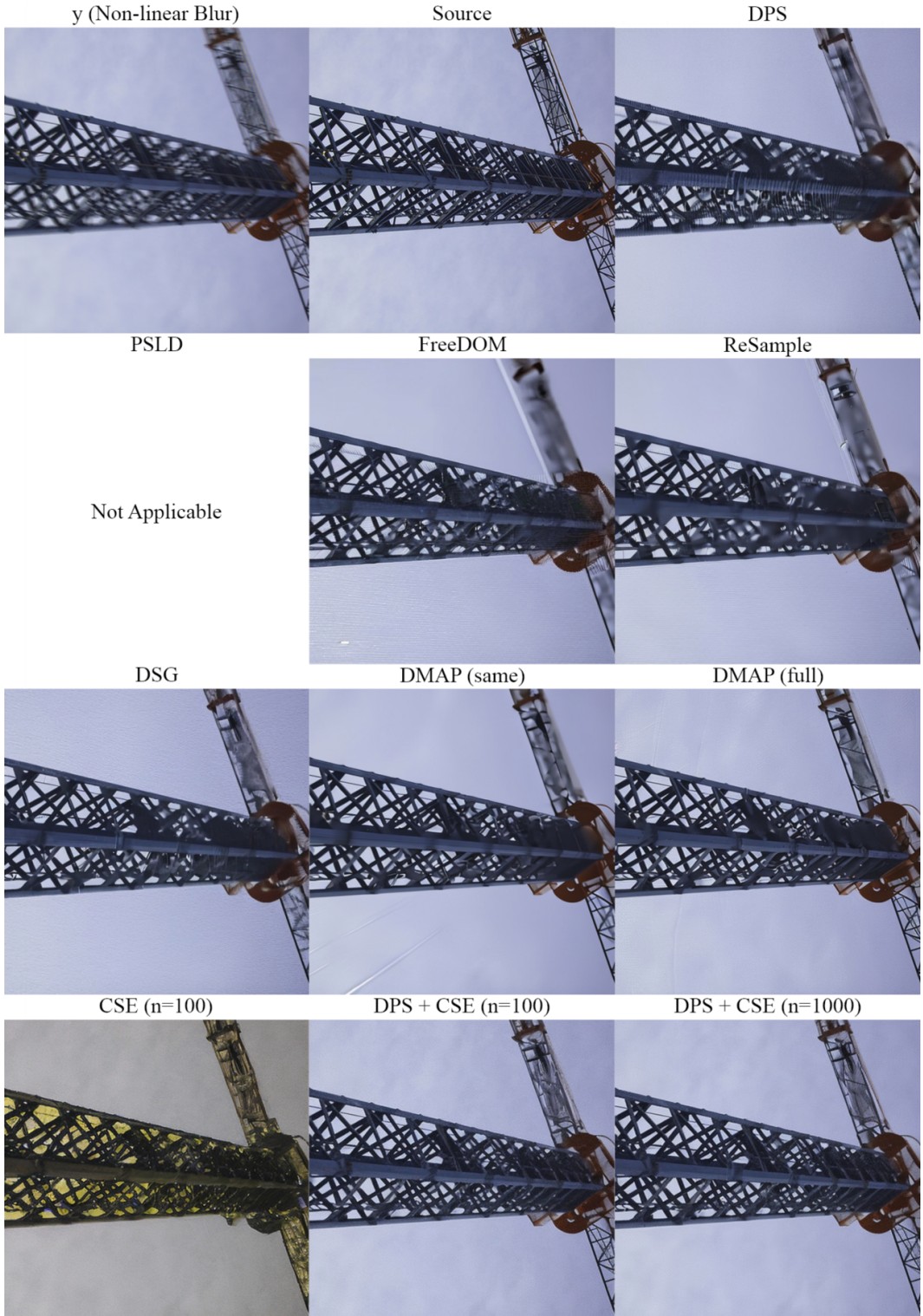

Figure 12: Additional qualitative results for non-linear deblurring.

