# OpenReview forum: "Rethinking Diffusion Posterior Sampling: From Conditional Score Estimator to Maximizing a Posterior"
_ICLR.cc/2025/Conference — ICLR 2025 Poster_

### Official Review · Reviewer_3ddK · 2024-10-24

**Soundness:** 3
**Presentation:** 2
**Contribution:** 3
**Rating:** 6
**Confidence:** 3

**Summary:**

The authors challenge the traditional view that DPS functions as a conditional score estimator, arguing instead that it aligns more closely with MAP. Through empirical analysis, they find that DPS exhibits significant score estimation errors, a high score mean, and low sample diversity, supporting this reinterpretation. To improve DPS, they propose explicitly maximizing the posterior using gradient ascent and introduce a lightweight conditional score estimator. These methods demonstrate improved results in terms of image quality, particularly for deblurring tasks.

**Strengths:**

1. The paper has the potential for significant impact as this is a popular field at the moment.
2. It introduces new ideas and reinterprets existing arguments, supported by data that validates this new perspective.
3. Building on their reinterpretation, the authors develop a new method that outperforms traditional DPS.

**Weaknesses:**

1. The paper has several grammatical flaws:
   - Captions for tables should be placed on top, as I believe this is required formatting by ICLR.
   - Add an arrow to indicate that lower FID is better, and do the same for other metrics.
   - There are many undefined acronyms, such as FID, KID, and LPIPS.
   - The captions for figures and tables could be improved for clarity, adding the takeaway from each table and image as in the text would benefit the reader.
   - In line 196, the notation for different math symbols is redundant with what is already explained on line 153.
   - The claim "With Observation I & II, one can conclude that DPS is far away from a reasonable conditional score estimator in practical scenarios" is quite extreme, especially since it is shown on just one problem.
   - In line 255, "we hypothesis" should be corrected to "we hypothesize."
   - In line 273, "(See Appendix C.1)" should not be placed after a sentence.
   - The first quotation marks throughout the paper are incorrect; in LaTeX, use `` for the opening quotation mark.
   - Citations like "Chung et al. (2022a); Song et al. (2023c)" are not grammatically correct and appear throughout the paper. Correct the citation style to improve readability.

2. There are no results for inpainting, unlike the original DPS paper.

**Questions:**

Is DPS not used for in painting as well? You seem to only provide results for deblurring.

---

> ### Author Response · Authors · 2024-11-17
>
> Thank you for your detailed review. We are pleased to provide answers to your questions:
> ## W1 The paper has several grammatical flaws
> Thank you for pointing out the grammatical flaws. We have corrected them in the updated paper.
>   * Captions should be on top: We have moved them to the top.
>   * Add an arrow to indicate that lower FID is better: We have included an arrow.
>   * There are many undefined acronyms: We have moved their definition to where they first appear.
>   * adding the takeaway from each table and image: We have included takeaways so readers can understand the content without having to refer to the explanations in the paper.
>   * the notation for different math symbols is redundant: We have removed redundant notations.
>   * The claim is quite extreme: We have modified it into: With Observation I & II, one can conclude that DPS is not an effective conditional score estimator for image restoration.
>   * "we hypothesis" should be corrected to "we hypothesize.": We have fixed it.
>   * "(See Appendix C.1)" should not be placed after a sentence.: We have fixed it.
>   * The first quotation marks throughout the paper are incorrect: We have corrected them.
>   * citation style: The citation command \citet{} of Natbib, which produces "Chung et al. (2022a); Song et al. (2023c)", is also used in the ICLR 2025 template (See https://github.com/ICLR/Master-Template/blob/05833d63fe48bbf250b144741ea77691018bb328/iclr2025/iclr2025_conference.tex#L163). Additionally, the way we create multiple citations using \citet{} aligns with the Natbib reference sheet (See Page 1 of https://gking.harvard.edu/files/natnotes2.pdf). Also, this citation style appears in outstanding papers of ICLR 2024 [Never Train from Scratch: Fair Comparison of Long-Sequence Models Requires Data-Driven Priors] and [Towards a statistical theory of data selection under weak supervision
> ]. Therefore, we believe this citation style is appropriate for ICLR.
>
> ## W2 and Q2: results for inpainting
> Thank you for your suggestions. Below we provide the results for the inpainting operator on 1000 ImageNet images. We used a fixed box inpainting mask: a 128x128 rectangle centered at a 512x512 image. We have demonstrated that for the inpainting operator, our two proposed improvements (DMAP & DPS + CSE) also significantly boost the performance of DPS.
>
>
> |   |Time(s)🡓 | PSNR🡑 | LPIPS🡓 | FID🡓 |
> |---|---|---|---|---|
> | DPS | 173 | 23.59 | 0.3090 | 53.28 |
> | PSLD | 230 | 23.87 | 0.3112 | 52.47 |
> | DSG | 193 | 23.53 | 0.3267 | 48.94 |
> | DMAP (same) | 187 | 24.37 | 0.2771 | 48.35 |
> | DMAP (full) | 484 | 24.59 | 0.2642 | 47.79 |
> | DPS + CSE (n=100)  | 189 + 28| 23.97 | 0.2482 | 31.51 |
> | DPS + CSE (Self-generated) | 189 + 28 | 24.01 | 0.2225 | 34.35 |

---

> > ### Comment · Reviewer_3ddK · 2024-11-25
> >
> > Thank you for addressing my comments. When referring to the citations being grammatically incorrect, in English, you cannot write:
> >
> > >  Chung et al. (2022a); Song et al. (2023c) propose that DPS works as a conditional score estimator, and provide an upper-bound on the estimation error.
> >
> > The semicolon is incorrect. One possible solution:
> >
> > > Chung et al. (2022a) and Song et al. (2023c) propose that DPS works as a conditional score estimator, and provide an upper-bound on the estimation error.
> >
> > I have increased my score accordingly but believe the authors should refine the writing of the paper.

---

> > > ### Author Response · Authors · 2024-11-26
> > >
> > > Thanks for the additional comments. We have updated the paper accordingly.

---

### Official Review · Reviewer_omzH · 2024-11-04

**Soundness:** 3
**Presentation:** 4
**Contribution:** 3
**Rating:** 8
**Confidence:** 3

**Summary:**

This paper challenges the prevailing view that Diffusion Posterior Sampling (DPS) works by approximating conditional score, hypothesizing that DPS is closer to maximizing a posterior (MAP) and proposes two improvements for DPS based on the new hypothesis.

**Strengths:**

* The paper presents compelling evidence through empirical analysis on 512x512 ImageNet images that DPS does not effectively approximate conditional score, contradicting prior understanding.
* The proposed improvements, explicit MAP implementation (DMAP) and a light-weighted conditional score estimator, are shown to significantly enhance the performance of DPS.
* The paper is well-organized and provides clear explanations of the technical details, including the mathematical formulations and algorithms.

**Weaknesses:**

* The paper mainly focuses on image restoration tasks and the generalizability of the MAP hypothesis to other inverse problems is not explored.
* While the light-weighted conditional score estimator is efficient, its performance still lags behind well-trained conditional diffusion models like StableSR.
* The authors claim that the MAP hypothesis can explain why Adam helps DPS, but this claim is not sufficiently substantiated.

**Questions:**

* To further investigate the hypothesis that DPS is closer to MAP than conditional score estimation, one suggestion is to experiment with toy distributions where the ground truth posterior is known. This approach would allow for a direct numerical comparison between the results of DPS and the true posterior.

---

> ### Author Response · Authors · 2024-11-18
>
> Thank you for your detailed review. We are pleased to provide answers to your questions:
> ## W1 the generalizability of the MAP hypothesis to other inverse problems is not explored
> Thanks for your suggestions. Below we provide the results for bedroom segmentation. we use 100 LSUN bedroom images and we are running 1000 images experiments. The segmentation model recognizes the layout of bedroom, as described in [Indoor Scene Layout Estimation from a Single Image]. To evaluate the segmentation result, we adopt a different set of metrics from image restoration, including MIoU and FID.
>
> In the table below, we have shown that for segmentation operator, our two proposed improvements (DMAP & DPS + CSE) also boost the performance of DPS significantly.
>
> |   |Time(s)🡓 | MIoU🡑 | FID🡓 |
> |---|---|---|---|
> | DPS | 184 | 0.3358 | 116.7 |
> | DSG | 185 | 0.3790 |114.7 |
> | DMAP (same) | 203 | 0.3883 | 102.7 |
> | DMAP (full) | 485 | 0.4188 |104.9 |
> | DPS + CSE (n=100)  | 185 + 28 | 0.3832 | 110.3 |
> | DPS + CSE (Self-generated) | 185 + 28 | 0.4159 | 107.8 |
>
> ## W2 its performance still lags behind well-trained conditional diffusion models like StableSR
> The objective of DPS and this paper is zero-shot (no training data) and few-shot (few training data) inverse problem solving. Despite the performance of DPS and this paper is worse than StableSR, the training cost of DPS and this paper is much smaller than StableSR. More specifically, DPS and DMAP (Improvement I in this paper) requires no training. DPS + CSE (Improvement II in this paper) requires 100-1000 images and 8 GPU hours training. On the other hand, StableSR requires 20,000 images and several days of training. In that sense, StableSR does not dominate DPS or this paper.
>
> The study of zero-shot and few-shot approaches is very important to diffusion model. We believe effective zero-shot and few-shot approaches can greatly broaden the application of pre-trained diffusion models.
>
> ## W3 why Adam helps DPS is not sufficiently substantiated
> Thanks for the suggestion, we will provide a more detailed explaination:
> * He et al. (2023) find that Adam improves the performance of DPS. This statement is not reasonable if DPS is a conditional score estimator, because Adam is an adaptive learning rate optimizer. In other words, Adam scales the score $\nabla_{X_{t}}\log p(y|X_t)$ unevenly in different dimensions. This scaling, theoretically, makes the estimation of $\nabla_{X_{t}}\log p(y|X_t)$ inaccurate, and should deteriorate the performance of DPS.
> *  However, if we assume DPS is solving the MAP in Eq. 16, then Adam, as a general method of gradient ascent, may not harm the performance of DPS. And this explains the phonemena that Adams improve DPS, observed by He et al. (2023).
>
> ## Q1 experiment with toy distributions
> Sure, we provide a toy example below.
> * The source distribution $p(X_0)$ is 2D 2-GMM (Gaussian Mixture Model). The centers are $(-1,-1),(+1,+1)$ and the diagonal $\sigma_0=0.3$.
> * The operator f is inpainting. More specifically, the second dimension of the random variable is set to 0. The measurement $y=(0.5,0)$
> * The diffusion is 100 steps karras diffusion with $\sigma_{max} = 4$ and $\rho=7$.
> * The diffusion scheduler is Euler Ancestral, and the DPS $\zeta_t=0.05$.
> The source distribution, true posterior and the DPS result is shown in this anonymous link: https://ibb.co/T2VSn32
> It is shown that the true posterior is a two mode distribution. However, the samples of DPS cover only one mode of true posterior. In that example, the behaviour of DPS is closer to MAP.

---

### Official Review · Reviewer_7J3w · 2024-11-04

**Soundness:** 3
**Presentation:** 3
**Contribution:** 3
**Rating:** 6
**Confidence:** 3

**Summary:**

This paper points out the ineffectiveness of Diffusion Posterior Sampling(DPS), revealing that DPS aligns more closely with maximum a posteriori(MAP) estimation rather than conditional score estimation. This paper demonstrates that DPS’s score estimation diverges from an accurate conditional model and produces low-diversity samples. The authors enhance DPS by explicitly maximizing the posterior with gradient ascent and incorporating a lightweight conditional score estimator.

**Strengths:**

- This paper is well organized. I enjoy reading this paper.
- This work suggests MAP hypothesis that can explain the recent mysterious observations regarding DPS. I think this is clear and novel contribution.
- Significant performance gain by being more faithful to MAP estimate well supports their hypothesis.
-

**Weaknesses:**

- Although I think insights from this paper is beneficial, this paper heavily relies on the empirical observations to develop and support their claims. It is okay, but theoretical insights will be strengthen their claim and more naturally lead to their MAP hypothesis.
- This work is limited to show that MAP hypothesis can explain the weird part of DPS rather than clarifying the reason why it works in MAP estimate.

**Questions:**

When distribution is multi-modal, is it possible to say that working as MAP estimate is the main reason for low-diversity generation?

---

> ### Author Response · Authors · 2024-11-17
> **Official Comment by Authors Part I**
>
> Thank you for your detailed review. We are pleased to provide answers to your questions:
>
> ## W1 ...theoretical insights will be strengthen their claim and more naturally lead to their MAP hypothesis...
>
> Currently, we have not figured out how to directly derive MAP from the original DPS (Chung et al., 2022a). To compensate for this, we additionally provide a toy example, and derive MAP from a variant of DPS.
>
> To better connect the original DPS and MAP, we provide a toy example below:
>   * The source distribution $p(X_0)$ is 2D 2-GMM (Gaussian Mixture Model). The centers are $(-1,-1),(+1,+1)$ with the diagonal $\sigma_0=0.3$.
>   * The operator $f(.)$ is inpainting. More specifically, the second dimension of the random variable is set to 0. And the measurement $y=(0.5,0)$
>   * The diffusion process uses 100 steps of Karras diffusion with $\sigma_{max} = 4$ and $\rho=7$.
>   * The diffusion scheduler is Euler Ancestral, and the original DPS $\zeta_t=0.05$.
> * The source distribution, true posterior and the original DPS result is shown in this anonymous link: https://ibb.co/T2VSn32
> * It is shown that the true posterior is a bimodal distribution. However, the samples from the original DPS cover only one mode of the true posterior. In this example, the behavior of the original DPS is closer to that of MAP.
>
> On the other hand, with additional assumptions, it is possible to derive MAP directly from a variant of DPS: DSG (Yang et al., 2024). More specifically, DSG improves DPS with a spherical projection. The update of DSG is as follows:
> $$
> X_{t-1} = \mathbb{E}[X_{t-1}|X_t] - \sqrt{n}\sigma_t\frac{\nabla_{X_t}||f(\mathbb{E}[X_0|X_t]) - y||}{||\nabla_{X_t}||f(\mathbb{E}[X_0|X_t]) - y||||},
> $$
> __Assumption 5.__
> Additionally, we need some assumptions:
> * 1.The posterior mean $\mathbb{E}[X_0|X_t]$ is a perfect approximation to posterior samples from $p(X_0|X_t)$. In that case, the approximation error in Eq. (7)(8) diminishes, i.e.,
>     $$
>     p_{\theta}(y|X_t) = p(y|X_0=\mathbb{E}[X_0|X_t]) = \exp{-\zeta_t||f(\mathbb{E}[X_0|X_t]) - y||}.
>     $$
> * 2.The $\sigma_t^2$ in Eq. (5) is so small that $\log p(y|X_{t-1})$ is locally linear in range of $\sqrt{n}\sigma_t$. The $X_t, X_{t-1}$ is so close that $\nabla_{X_{t-1}}\log p(y|X_{t-1}) \approx \nabla_{X_t}\log p(y|X_{t})$.
>
> Under those assumptions, we can show that DSG is also a MAP:
> __Proposition 6.__ The DSG solves the MAP problem $X_{t-1} \leftarrow \arg\max p_{\theta}(X_{t-1}|X_t,y)$ in __Hypothesis 2__ when dimension $n$ is high.
> * __proof__: We first consider a gradient ascent with learning rate $\alpha_t$:
>     $$
>     X_{t-1} = \mathbb{E}[X_{t-1}|X_t] + \alpha_t \nabla_{X_t}\log p(y|X_t).
>     $$
>     As we have assumed in __Assumption 5.2__, the $\log p(y|X_t)$ is locally linear and $\nabla_{X_{t-1}}\log p(y|X_{t-1}) \approx \nabla_{X_t}\log p(y|X_{t})$. As gradient is the direct of steepest descent, when $\alpha_t$ changes, the $X_{t-1} = \mathbb{E}[X_{t-1}|X_t] + \alpha_t \nabla_{X_t}\log p(y|X_t)$ is the maximizer of $\log p(y|X_{t-1})$, given the distance from the point $X_{t-1}$ to center $\mathbb{E}[X_{t-1}|X_t]$ is fixed.
>     On the other hand, the DSG can be written as the above gradient ascent
>     $$
>     X_{t-1} =  \mathbb{E}[X_{t-1}|X_t] - \sqrt{n}\sigma_t\frac{\nabla_{X_t}||f(\mathbb{E}[X_0|X_t]) - y||}{||\nabla_{X_t}||f(\mathbb{E}[X_0|X_t]) - y||||} \\
>     \overset{a}{=} \mathbb{E}[X_{t-1}|X_t] + \frac{\sqrt{n}\sigma_t}{||\nabla_{X_t}f(\mathbb{E}[X_0|X_t]) - y||||}\nabla_{X_t}\log p(y|X_{t}).
>     $$
>     where (a) holds due to __Assumption 5.1__. We further notice that DSG is equivalent to the above gradient ascent. Further, from the update procedure of DSG, the distance of $X_{t-1}$ to $\mathbb{E}[X_0|X_t]$ is $\sqrt{n}\sigma_t$. Therefore, the result of DSG is the minimizer of $\nabla_{X_{t-1}}p(y|X_{t-1})$, on the sphere surface centered at $\mathbb{E}[X_{t-1}|X_t]$ with radius $\sqrt{n}\sigma_t$:
>     $$
>     X_{t-1} \leftarrow \arg\max \log p(y|X_{t-1}), X_{t-1} \in \mathcal{S}(\mathbb{E}[X_{t-1}|X_t],\sqrt{n}\sigma_t),
>     $$
>     where $\mathcal{S}(\mathbb{E}[X_{t-1}|X_t],\sqrt{n}\sigma_t)$ is the sphere surface centered at $\mathbb{E}[X_{t-1}|X_t]$ with radius $\sqrt{n}\sigma_t$. Besides, from (Yang et al., 2024), we know that as dimension $n\rightarrow \infty$, $X_{t-1}\sim p(X_{t-1}|X_t)$ is equivalent to $X_{t-1}\in \mathcal{S}(\mathbb{E}[X_{t-1}|X_t],\sqrt{n}\sigma_t)$. Then as dimension $n\rightarrow \infty$, DSG+DPS is eqvalient to:
>     $$
>     X_{t-1} \leftarrow \arg\max \log p(y|X_{t-1}), X_{t-1}~p(X_{t-1}|X_t),
>     $$
>     which is again equivalent to __Hypothesis 2__ by __Proposition 3__. $\square$

---

> ### Author Response · Authors · 2024-11-17
> **Official Comment by Authors Part II**
>
> Despite DPS+DGS being already MAP under __Assumption 5__, our method remains to be MAP without it. Therefore, DSG alone can not replace DMAP. And practically, its performance is inferior to DMAP as shown in Table 4.
>
> With __Assumption 5.1__, the original DPS is equivalent to classifier-based guidance (Dhariwal 2021). However, what is classifier-based guidance on earth theoretically is not yet well explored. Therefore, we leave the strict theoretical relationship between original DPS and MAP to future works. Once this problem is solved, the nature of classifier-based guidance can be revealed. We believe this problem is beyond the scope of a paper focused on DPS.
>
> ## W2 ...clarifying the reason why it works in MAP estimate...
> We can think DPS as a simplification of DMAP, with one step gradient ascent and without spherical projection. As long as the step size of diffusion model is small, the $\log p(y|X_{t-1})$ can be locally linear (__Assumption 5.2__). In that case, one step of gradient ascent is enough. As long as the DPS guidance step is much smaller than the DDPM step, the distance from $X_{t-1}$ to the sphere surface $\mathcal{S}(\mathbb{E}[X_{t-1}|X_t],\sqrt{n}\sigma_t)$ is small. In that case, spherical projection is no longer necessary. And when those conditions are approximately satisfied, DPS also works well.
>
> ## Q1 ...MAP estimate is the main reason for low-diversity generation..
> Yes. The MAP estimate always go to the highest mode. When the distribution is multi-modal, other smaller modes will be neglected by MAP. And therefore, MAP causes low-diversity generation. This phenomena is also shown in the toy example.

---

> > ### Comment · Reviewer_7J3w · 2024-11-27
> >
> > Thanks for your answer. I am happy to keep my score

---

### Author Response · Authors · 2024-11-19
**Summary of Revisions**

Thank you for your detailed review. We have uploaded the revised paper, with all the revisions marked in blue. Below is a summary of revisions:
* We include the result of inpainting and segmentation operators, as suggested by 3ddK and omzH.
* We include the toy example showing DPS is closer to MAP, as suggested by omzH and 7J3W.
* We include additional theoretical discussions on the relationship between DPS and MAP, and why DPS works under MAP assumption, as suggested by 7J3W.
* We expand discussions on why Adam helps DPS, as suggested by omzH.
* We fix the grammar errors, as suggested by 3ddK.

We hope that we have addressed your concerns and we are glad to provide additional clarifications.

---

### Meta-Review · Area_Chair_QE8G · 2024-12-20

**Metareview:**

This paper revisits the principles underlying Diffusion Posterior Sampling (DPS), proposing that it aligns more closely with a maximum a posteriori (MAP) estimation rather than a conditional score estimator. The authors provide a comprehensive empirical analysis, demonstrating the limitations of DPS’s conditional score estimation and its low sample diversity. Building on their findings, the paper introduces two improvements: explicitly maximizing the posterior (DMAP) and incorporating a lightweight conditional score estimator (CSE). These methods show substantial performance gains across multiple tasks, including image restoration, segmentation, and inpainting.

**Additional Comments On Reviewer Discussion:**

Reviewers found the paper clear and well-organized. Its strengths include the novel reinterpretation of DPS through the MAP framework and the effectiveness of the proposed enhancements. While some concerns were raised about limited theoretical grounding and generalizability beyond image restoration, the authors addressed these issues by providing additional experiments and clarifications. The empirical results, particularly on challenging datasets like 512×512 ImageNet, support the paper's claims, making it a valuable contribution.

---

### Decision · Program_Chairs · 2025-01-22

Accept (Poster)